# Universal, untargeted detection of bacteria in tissues using metabolomics workflows

Wei Chen[1], Min Qiu[1], Petra Paizs[2], Miriam Sadowski[3], Toma Ramonaite[2], Lieby Zborovsky[1], Raquel Mejias-Luque [4], Klaus-Peter Janßen [5], James Kinross [6], Robert D. Goldin [2], Monica Rebec[7], Manuel Liebeke [3,8], Zoltan Takats [2,9], James S. McKenzie [2,10] ✉ & Nicole Strittmatter [1,10] ✉

Fast and reliable identification of bacteria directly in clinical samples is a critical factor in clinical microbiological diagnostics. Current approaches require time-consuming bacterial isolation and enrichment procedures, delaying stratified treatment. Here, we describe a biomarker-based strategy that utilises bacterial small molecular metabolites and lipids for direct detection of bacteria in complex samples using mass spectrometry (MS). A spectral metabolic library of 233 bacterial species is mined for markers showing specificity at different phylogenetic levels. Using a univariate statistical analysis method, we determine 359 so-called taxon-specific markers (TSMs). We apply these TSMs to the in situ detection of bacteria using healthy and cancerous gastrointestinal tissues as well as faecal samples. To demonstrate the MS method-agnostic nature, samples are analysed using spatial metabolomics and traditional bulk-based metabolomics approaches. In this work, TSMs are found in >90% of samples, suggesting the general applicability of this workflow to detect bacterial presence with standard MS-based analytical methods.

Rapid and reliable detection and identification of bacteria are crucial for clinical diagnosis, personalised medicine, and prevention of treatment resistance[1]. Traditional, culturing-based microbiology methods relying on morphology and/or biochemical tests are both time-consuming and labour-intensive, taking up to 5–7 days for certain pathogenic microorganisms[2–4]. Additionally, these techniques are limited by the availability of commercial culturing media to detect unknown and difficult-to-cultivate bacterial species[2,5]. In recent years, matrix-assisted laser desorption/ionisation-mass spectrometry (MALDI-MS) has been rapidly adopted in clinical practice to identify bacteria based on unique protein profiles[6,7]. However, while the

identification process itself is faster, MALDI-MS-based analysis suffers from the same time-consuming isolation step as traditional approaches. Time, however, is a decisive factor in the successful treatment of several infection scenarios such as sepsis[4,8]. Consequently, the ideal scenario of microbial diagnosis is to identify bacteria directly from the clinical sample. Using MALDI, so far only a small number of applications could be implemented due to the high protein background in most clinical samples[9]. These assays typically require a combination of extraction and centrifugation steps[10,11], nanoparticle or microfluidic techniques for isolating and enriching bacteria from complex biofluid samples (e.g., blood, saliva, and cerebrospinal fluid)[9,12,13].

[1]Department of Bioscience, School of Natural Sciences, Technical University of Munich, Garching, Germany. [2]Department of Metabolism, Digestion and Reproduction, Imperial College London, London, United Kingdom. [3]Department of Symbiosis, Max Planck Institute for Marine Microbiology, Bremen, Germany. [4]Institute for Medical Microbiology, Immunology and Hygiene, School of Medicine and Health, Technical University of Munich, Munich, Germany. [5]Department of Surgery, School of Medicine and Health, Technical University of Munich, Munich, Germany. [6]Department of Surgery and Cancer, Imperial College London, London, United Kingdom. [7]North West London Pathology, Imperial College Healthcare NHS Trust, London, United Kingdom. [8]Department for Metabolomics, Institute for Human Nutrition and Food Science, University of Kiel, Kiel, Germany. [9]Department of Immunomedicine, University of Regensburg, Regensburg, Germany. [10]These authors contributed equally: James S. McKenzie, Nicole Strittmatter.
✉e-mail: j.mckenzie@imperial.ac.uk; nicole.strittmatter@tum.de

Metabolomics is the comprehensive study of small molecules (metabolites) within cells, biofluids, tissues, or organisms. These metabolites are the final downstream products of cellular processes and provide a snapshot of the physiological state of an organism at a given time. By analysing metabolomic patterns, we can identify specific biomarkers that differentiate between biological states, making metabolomics a powerful tool for identification of different disease states and understanding host-pathogen interactions[14]. Bacterial lipids and small molecule metabolites also pose an alternative route for direct bacterial identification[15,16]. This approach was used most notably in the MIDI Microbial Identification System (MIDI, Inc. Newark, Delaware, USA) using pyrolysis-gas chromatography-MS to profile bacterial fatty acid patterns[16–20]. While biotyping using MALDI-MS has dominated the clinical diagnostic field in recent years, research into small-metabolite patterns of microorganisms continued using different, predominantly ambient, MS techniques. These patterns have often proved to be similarly specific in characterising pure cultures and phenotyping even closely related bacterial species[21–33]. Notable examples include rapid evaporative ionisation-mass spectrometry (REIMS), nano-DESI, and the MasSpec Pen.

In previous studies, it was proposed regularly that species-specific biomarkers can be used to detect bacteria in situ in clinical specimens. Bregy et al. suggested the concept of volatile bacterial biomarkers for the detection of periodontal pathogens[22]. Using secondary electrospray ionisation-MS, the authors first performed untargeted headspace analysis of the in vitro cultures of periodontal pathogens *Aggregatibacter actinomycetemcomitans*, *Porphyromonas gingivalis*, *Tanerella forsythia,* and *Treponema denticola*. They detected 120 bacteria-specific compounds and subsequently used these to detect bacteria in the saliva of a periodontitis patient ($n = 1$) and controls ($n = 2$). 18 compounds from all four species were found to increase in the periodontitis specimen. In 2022, Povilaitis et al. used the MassSpec Pen to detect the causative agent in a number of methicillin-resistant *Staphylococcus aureus* (MRSA)-positive clinical specimens, including synovial fluid, sacral tissue, and tissue smears[24]. Lasso classifiers were built on Gram-level, *Staphylococcus vs. Streptococcus* level, and *S. aureus vs S. epidermis* level, which involved glycerophospholipids (GPL) and quorum-sensing molecules as predictive features. Whereas all pure cultures isolated from these samples were correctly identified, only the Gram classification worked reliably in clinical specimens. Jarmusch et al. have proposed an approach deploying touch spray ionisation (TSI)-MS in which a medical-grade swab that can directly emit an electrospray was developed and shown to be able to differentiate between *Streptococcus pyogenes* and *Streptococcus agalactiae*

based on spectral fingerprints[26]. The authors suggested using this device to detect *S. pyogenes* on throat swabs for the clinical diagnosis of streptococcal throat, proving that bacterial lipids can be detected off a simulated throat swab specimen. Further promising applications include the use of an automated, rotation stage DESI approach to assess the mucosal microbiome directly in vaginal swab specimens[27,28] or using REIMS to assess microbial communities in cystic fibrosis samples of respiratory and non-respiratory specimens such as blood cultures, urine, drain fluid, vaginal, and skin swab samples[21].

There are however four challenges associated with the direct detection of bacteria from clinical specimens that remain to be evaluated systematically: (i) the range of possible pathogenic agents is much larger (~1400 pathogenic species are known)[34] than any of the previously used datasets deployed for training purposes which usually range from 6 to 25 bacterial species only[33,35–38]; (ii) bacterial markers extracted from such undersized datasets may not be as specific or conserved as initially thought; (iii) the variation of bacterial cell numbers varies widely in different infection scenarios[35,39,40]; (iv) most of the possible clinical specimen types (e.g., saliva, mucosal lining fluid) are typically not sterile[40], leading to a range of commensal organisms to be expected alongside the pathogen.

In this work, we address especially points (i) and (ii) to assess the feasibility of a biomarker-based strategy that includes conserved small-metabolite-based markers capable of identifying specific bacterial taxa (from phylum to species), which we refer to as taxon-specific markers (TSMs). 359 TSMs were found and subsequently deployed to identify pure bacterial cultures. We further touch upon points (iii) and (iv) by assessing whether these markers can be used to detect bacteria in situ in complex matrices.

## Results

### Feasibility of taxon-specific markers

A spectral database was constructed from bacterial profiles recorded using REIMS, containing 3274 individual bacterial strains across 233 different bacterial species. From these, a smaller training database was created that comprises spectral profiles of 597 of these 3274 individual strains, including every unique species and limiting the overall number of entries per species to a maximum of seven. A representation of the composition of the training dataset is shown in Fig. 1. These 233 species belong to 81 genera, 47 families, 23 orders, 11 classes, and 5 phyla of opportunistic pathogens and commensals. 22 groups were identified to genus-level only (such as "*Comamonas* sp."). As these were frequently rare isolates, they were included in the database for better coverage of the clinically relevant bacterial pool.

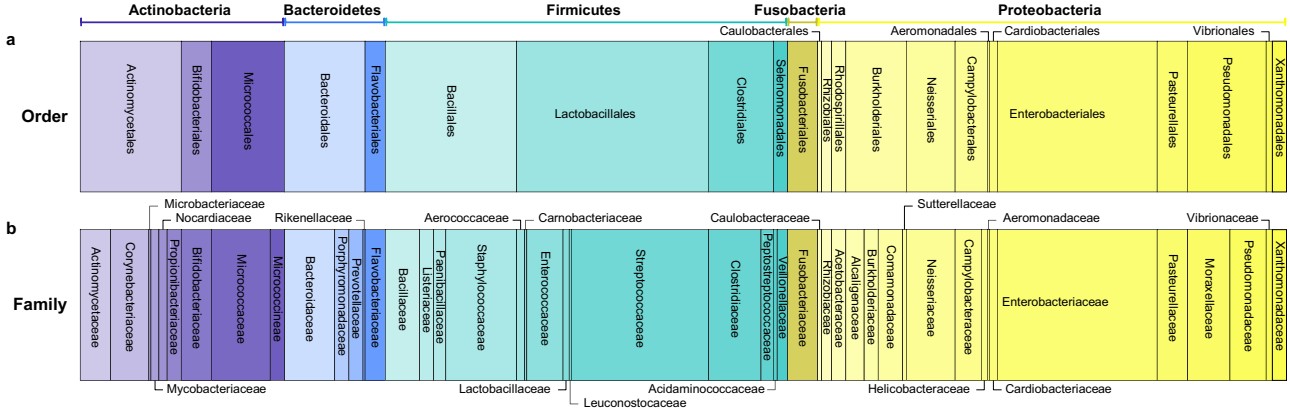

**Fig. 1 | Overview training database composition.** Horizontal stacked bar representation of order- (**a**) and family-level (**b**) of the database used to derive TSMs. The width of each bar is related to the percentage coverage of the database comprising $n = 597$ bacterial profiles of 233 bacterial species. Source data are provided as a Source Data file. Phylum level is indicated using brackets and colour (purple: Actinobacteria, blue: Bacteroidetes, cyan: Firmicutes, green-yellow: Fusobacteria, yellow: Proteobacteria). A detailed table on database composition can be found in Supplementary Data 1.

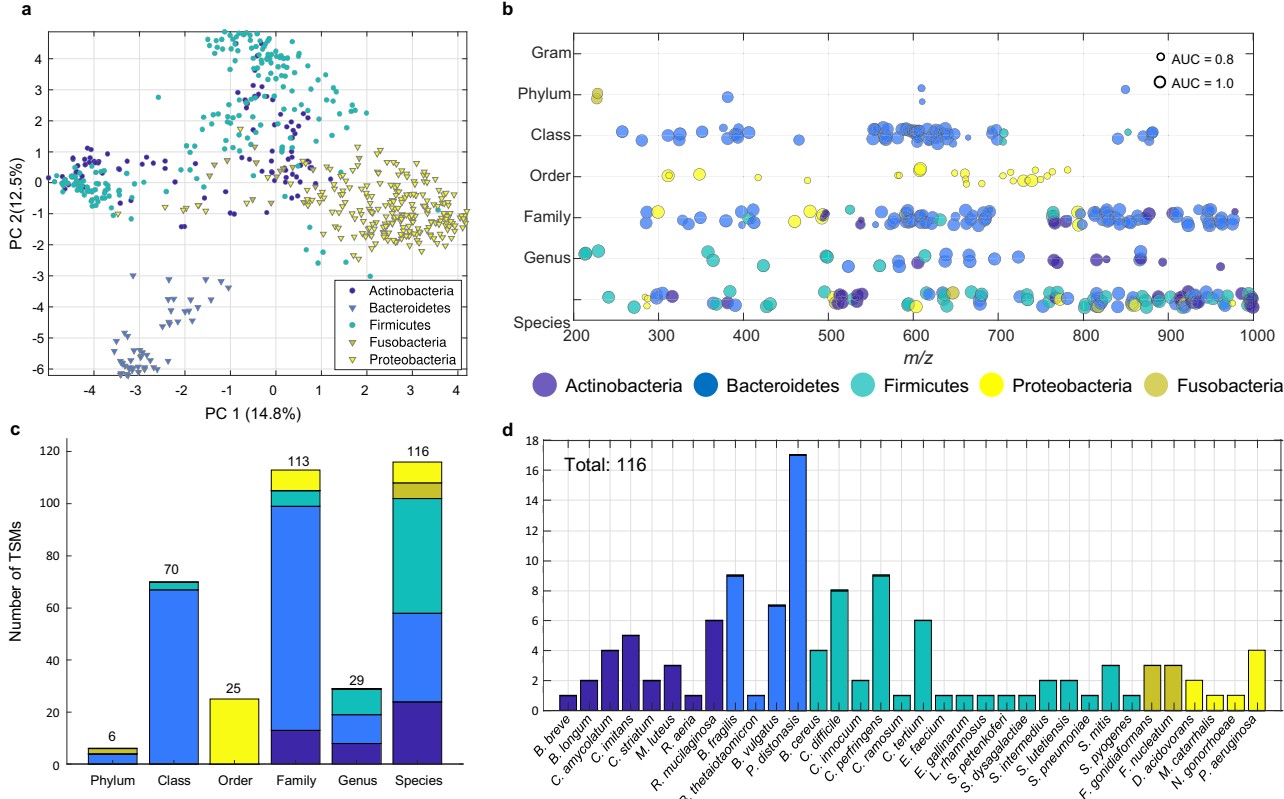

**Fig. 2 | Summary of taxon-specific markers. a** PCA scores plot of the dataset used to derive TSMs ($n = 597$ isolates belonging to 233 bacterial species). Circles represent Gram-positive, triangles represent Gram-negative species. The percentage of variance explained by PCs 1 and 2 are given in the axes labels. **b** Representation of all TSMs ($n = 359$) found on different phylogenetic (taxonomic) levels as a function of $m/z$ value. Circle size indicates the area under the curve (AUC) from the receiver operating characteristic (ROC) curve. A detailed list including p- and AUC values

can be found in Supplementary Data 2. **c** Summary of the number of TSMs by phylogenetic level. **d** Summary of TSMs found on species level. All panels are coloured by phylum (purple: Actinobacteria, blue: Bacteroidetes, petrol: Firmicutes, yellow: Proteobacteria, green: Fusobacteria). A further breakdown of markers on each phylogenetic level can be found in Supplementary Fig. 1. Source data are provided as a Source Data file.

After spectral pre-processing, a total of 5866 unique $m/z$ values were detected over the entire training dataset comprising the metabolites from all 233 bacterial species. We have restricted our analyses to $m/z$ values from 200 to 1000 Da, as peaks with a higher and lower $m/z$ value generally showed low spectral intensity which complicates their detection. As a next step, we aimed to derive spectral features which demonstrated phylogenetic specificity. For this, we assessed the specificity of each of the 5866 spectral features at the seven different phylogenetic levels between Gram (negative/positive) and species levels. Using a univariate approach, a total of 359/5866 features were found to be specific at a certain phylogenetic level (see Fig. 2c) and are considered TSMs; the relevant criteria are explained in Materials and Methods.

Figure 2a shows the principal component (PC) scores, coloured according to the phylum of each of the 597 bacterial profiles in the training set. Separation across PCs 1 and 2 can be observed between Gram-positive and -negative bacteria; despite this, there were no features deemed to be suitably significant at discriminating at Gram-level.

TSMs can be found from phylum-level onwards, with 6, 70, 25, 113, 29, and 116 TSM detected on phylum-, class-, order-, family-, genus-, and species-level, respectively. The list of all TSMs can be found in Supplementary Data 2. As Fig. 2c (and Supplementary Fig. 1 as a more detailed breakdown) shows, most of these markers are found for the members of the phylum Bacteroidetes, although they only constitute <10% of database spectral profiles (see Fig. 1), suggesting a distinctly different metabolism in this phylum of bacteria. An exception is the level order, which is dominated by Rhodospirillales. This is likely due

to this group not being represented well within the dataset while being very homogenous, comprising only two species of the genus Roseomonas. 113 markers were found on the family level, suggesting that at this level phenotypic changes are large. Again, the majority (>80%) of these 113 TSMs were found to originate from families within the phylum Bacteroidetes.

29 TSMs were found on the genus level, equally originating from Actinobacteria, Firmicutes, and Bacteroidetes. The highest number of TSMs was found on the species level, with 116 markers originating from 34 bacterial species (see Fig. 2d). This corresponds to 36% of the bacterial species eligible for testing (those with at least 3 bacterial profiles/observations in the training set, see Supplementary Fig. 2). The largest individual number of markers were detected for the species *Parabacteroides distonasis* with >15 individual TSMs, while 5–9 markers were detected each for *Bacteroides fragilis, Bacteroides vulgatus, Clostridium difficile, Clostridium perfringens, Clostridium tertium, Corynebacterium imitans,* and *Rothia mucilaginosa.* Of the 116 species-specific TSMs, 43 are identified as isotope peaks. We decided to include these among TSMs as they can serve for corroborative purposes or as alternative markers in cases where there may be spectral interference in either the complex matrix or the training dataset. For instance, in 15 of these cases, the original peak is not among the TSM marker list, possibly due to overlap with spectral profiles of other bacterial species.

Using exact mass, isotopic patterns, database search (The Human Microbial Metabolome Database[41], *Pseudomonas aeruginosa* metabolite database[42], LIPID MAPS[43]), and literature search, we could

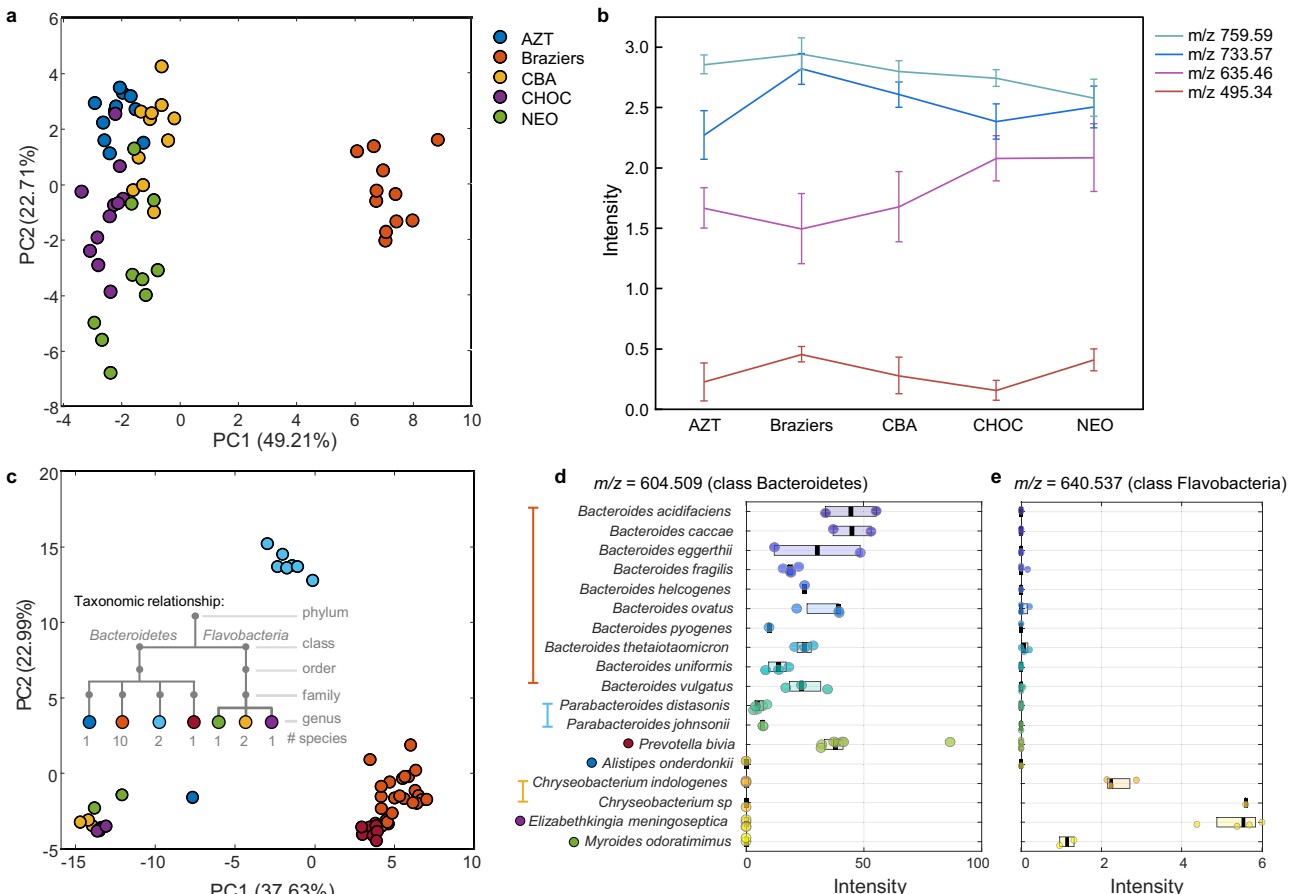

**Fig. 3 | Assessment of the quantitative nature of TSMs. a** PCA scores plot of data recorded for *Clostridium difficile* cultured on five different culturing media (Braziers: *C. difficile* selective medium (*n* = 11 isolates), CBA: Columbia blood agar (*n* = 11 isolates), CHOC: chocolate agar (*n* = 11 isolates), AZT: CBA with aztreonam [8 mg/L] for fastidious Gram-positives (*n* = 11 isolates), NEO: FAA with Neomycin for fastidious anaerobes (*n* = 10 isolates)). **b** Normalised intensities of TSMs found for *C. difficile* over the different culturing media. Data are presented as mean value ± standard deviation (SD). Intensities of TSM were calculated with ±15 ppm mass tolerance and normalised to the total intensity over the 200–1000 *m/z*

range followed by log$_{10}$ transformation. **c** PCA scores plot of data recorded for the Bacteroidetes phylum including taxonomic relationship. The colour code is shown beside the list of species in **d**. **d**, **e** Abundances of TSMs within species of Bacteroidetes phylum for class Bacteroidetes-specific marker at *m/z* 604.509 and class Flavobacteria-specific marker at *m/z* 640.537, respectively (*n* = 1–7 isolates, detailed information can be found in the Source Data). Box limits show the 25th and 75th percentiles, with the median marked in black. Intensities of TSM were calculated with ±15 ppm mass tolerance and normalised to the total intensity over the 200–1000 *m/z* range. Source data are provided as a Source Data file.

tentatively identify 177 of the overall 359 TSMs which are listed in Supplementary Data 3; no annotations were found for the remaining TSMs, which is associated with the only poorly characterised metabolic diversity among bacteria[44,45]. While not critically important for assessing the feasibility of bacterial small molecular biomarkers in general, precise metabolite annotation will eventually facilitate the translation of TSMs to other MS modalities by enabling the calculation of adduct exact masses under different experimental conditions and polarities.

**Ability of TSMs to quantify bacteria**
Quantitation can be performed on the level of the TSM marker itself or on the level of bacterial numbers. While the former is possible, given suitable standards are available, and the marker concentration lies in the linear range of the instrument, more factors need to be considered to quantify bacteria. Ideally, TSMs should be produced by bacteria in equal amounts irrespective of the growth conditions. To investigate the effect of different culturing conditions on TSM expression, we have cultured 12 clinical isolates of *C. difficile* on five different rich culturing media. We observed a clear separation based on culturing media in the resulting PCA scores plot (see Fig. 3a).

Eight TSMs were found for *C. difficile* at the species level (5 excluding isotopes). The abundance of all species-level TSMs for *C. difficile* is observed to vary among the different culturing media (Fig. 3b). This highlights the requirement to assess the dependency on culturing media for each of the 359 TSMs under different nutrient-rich and -poor culturing conditions, to determine whether any of the markers have the potential to be used for direct quantification of bacteria.

Similar observations are made about marker abundance within different groups on the same taxon (e.g., different species on genus-level). Examples are shown in Fig. 3c–e for the phylum Bacteroidetes. While expression might be conserved among a single bacterial species, these levels frequently differ between distinct members of the same phylogenetic level. This is exemplified by *m/z* 640.537 (Fig. 3e), which shows expression among all members of the Flavobacteria, yet the abundance among the different species varies considerably. TSM with *m/z* 604.509 (Fig. 3d) is specific for the class of Bacteroidetes and assigned as ceramide 35:0;O3 as [M+Cl]⁻ adduct[46,47]. We have found this marker among all members of this class with the exception of Rikenellaceae. However, it must be noted that this family is only represented by a single file (*Alistipes oderdonkii*) and it is possible this

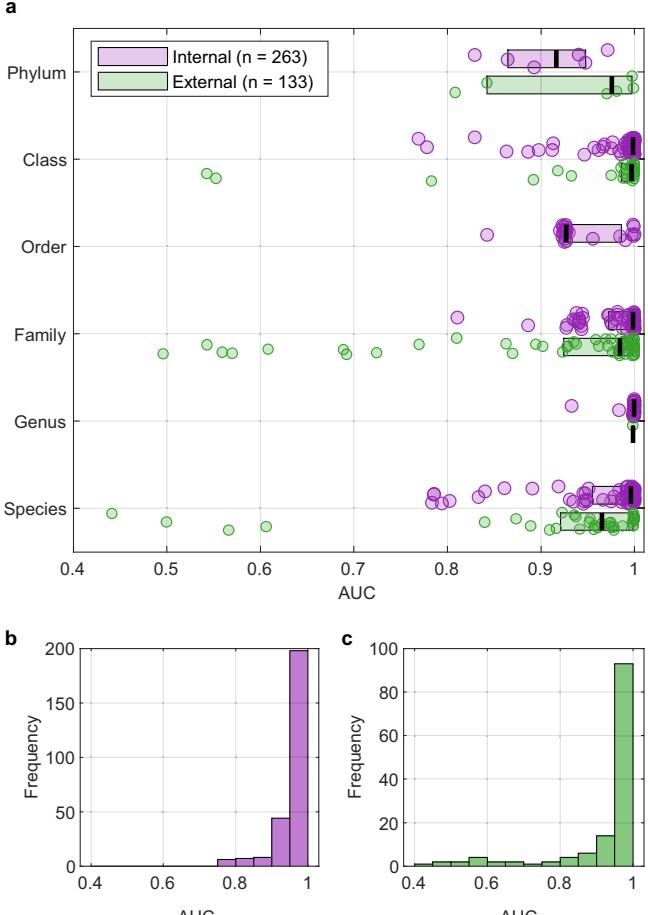

**Fig. 4 | Validation of TSMs using two validation datasets.** Area under the curves (AUCs) were determined for each variable where there were at least 3 bacterial profiles/observations in each dataset. **a** Boxplots showing AUCs for each taxon, with overlaid data points, for internal ($n = 263$ TSM) and external ($n = 134$ TSM) datasets. Box limits show the 25th and 75th percentiles, with the median marked in black. No order-level TSMs could be tested in the external dataset, and only one at the genus-level. Histograms of AUCs for **b** internal and **c** external datasets, spanning the range 0.4–1.0 in intervals of 0.05. The median values (and interquartile ranges) for internal and external markers are 0.997 (IQR 0.047) and 0.985 (IQR 0.071). IQR: interquartile range. Source data are provided as a Source Data file.

database entry might have been misidentified, further suggested by its clustering with the class of Flavobacteria in a PCA (see Fig. 3c, dark blue). This highlights the importance of parallel confirmation using sequencing-based analysis for future studies in order to further tighten the criteria for TSM selection.

A final aspect to consider is whether the produced markers are only intracellularly located (direct marker) or bacteria will secrete these compounds into their extracellular surroundings (indirect marker). An assessment of the localisation of markers has to be included in future database building. As the current database is built from solid agar cultures only, a distinction between intracellular and extracellular metabolites cannot be made.

### Validation of biomarkers on pure bacterial cultures
Testing whether TSMs can be used to identify pure bacterial cultures was performed using two datasets. The 'internal' set consists of the 2677 bacterial profiles from bacterial isolates that were excluded from the training set ($n = 597$ profiles) used to derive TSMs. The 'external' set contains 564 spectral profiles recorded on another independent cohort of clinical isolates on an Xevo G2-XS equipped with a REIMS

atmospheric pressure interface[48]. Peak intensities for the 359 variables were extracted from all files, using the same methodology as for the training database. As a consequence of the compositions of the validation sets, not all TSMs were tested due to having fewer than three bacterial profiles in the dataset. In the internal dataset AUCs for 263 variables were determined, while for the external dataset this number is 134. These results are summarised in Fig. 4 as both boxplots and histograms. The median (interquartile range, IQR) AUC values for internal and external variables are 0.997 (0.047) and 0.985 (0.071), respectively. The results show that many of these features are highly specific when using the same instrumental platform (unmodified Thermo Exactive classic), and only moderately lower when using an entirely different instrumental setup.

### TSM detectability directly in complex samples
To detect bacteria in complex biological specimens beyond those recorded using REIMS, it is important that these markers can be detected in an MS method-agnostic fashion. DESI-MS is the most widely used ambient ionisation technique and a particularly promising tool for direct on-sample applications such as swab analysis[27,28]. It is further one of the most widely used methods for tissue imaging[49–51]. A comparison of the spectral profiles for *B. fragilis* shows good general overlap of spectral features between DESI and REIMS, although individual peak intensities might differ (see Fig. 5). The majority of TSMs detected for *B. fragilis* and higher levels were detected successfully using DESI.

To demonstrate the applicability of the TSMs for the detection of bacteria in complex biological samples, we attempted to visualise the presence and distribution of bacteria in 44 human colorectal tissues recorded using DESI-MSI (see Supplementary Data 4 for access links). Tissue specimens were either from a tumour lesion or 5 to 10 cm distant from the lesion. Bacteria cover the mucosal membranes in the gut and the effect of the gut microbial community on health and disease has been abundantly proven[36,52].

Among the cancerous specimens, bacteria were largely found localised in necrotic regions or among lymphatic aggregates (see Supplementary Figs. 3 and 4). However, bacteria were also frequently detected along healthy mucosa (see Supplementary Fig. 5). Bacteria of at least one phylum could be visualised in 97.7% of the 44 analysed colorectal specimens both including healthy and cancerous tissues. No bacterial markers were detected in 1/44 samples (A15 10 cm S3, full sample list can be found in Supplementary Data 4). For the class Bacteroidetes, in total 81 (out of a total number of 143) TSMs were found over 41 samples. For other Gram-negative bacteria, Proteobacteria can be detected through 20 TSMs, while Fusobacteria were presented by three markers. Among Gram-positive phyla, Actinobacteria and Firmicutes can be detected by 17 and 22 markers, respectively.

Sequencing-based data of the same tissues are available (Supplementary Data 5), however, tissues were halved to enable both analyses, and thus the data is not strictly comparable. DESI-MSI analysis of TSM clearly shows spatial heterogeneity in bacterial composition and localisation into hotspots (Supplementary Figs. 3–5), which complicates the comparison of data obtained from two non-identical areas of tissues even within the same macroscopic tissue specimen. Thus, sequencing-derived results are only included for validation of the presence of a certain taxon in question.

DESI follows a similar ionisation mechanism as ESI, which is the standard ionisation interface in LC-MS analysis[53]. Consequently, spectra obtained from lipid extracts using both techniques show high similarity[38]. A correlation coefficient of 0.70 ($p < 0.001$) has been reported between the mass spectra of DESI-MS and LC-ESI-MS for GPL in the mass range of 600–1000 Da under negative mode[54]. To show that TSMs can also be detected using LC-MS datasets, we have screened an additional set of 48 faecal samples (unrelated to the specimen used for DESI-MSI) for TSMs. As currently no systematic

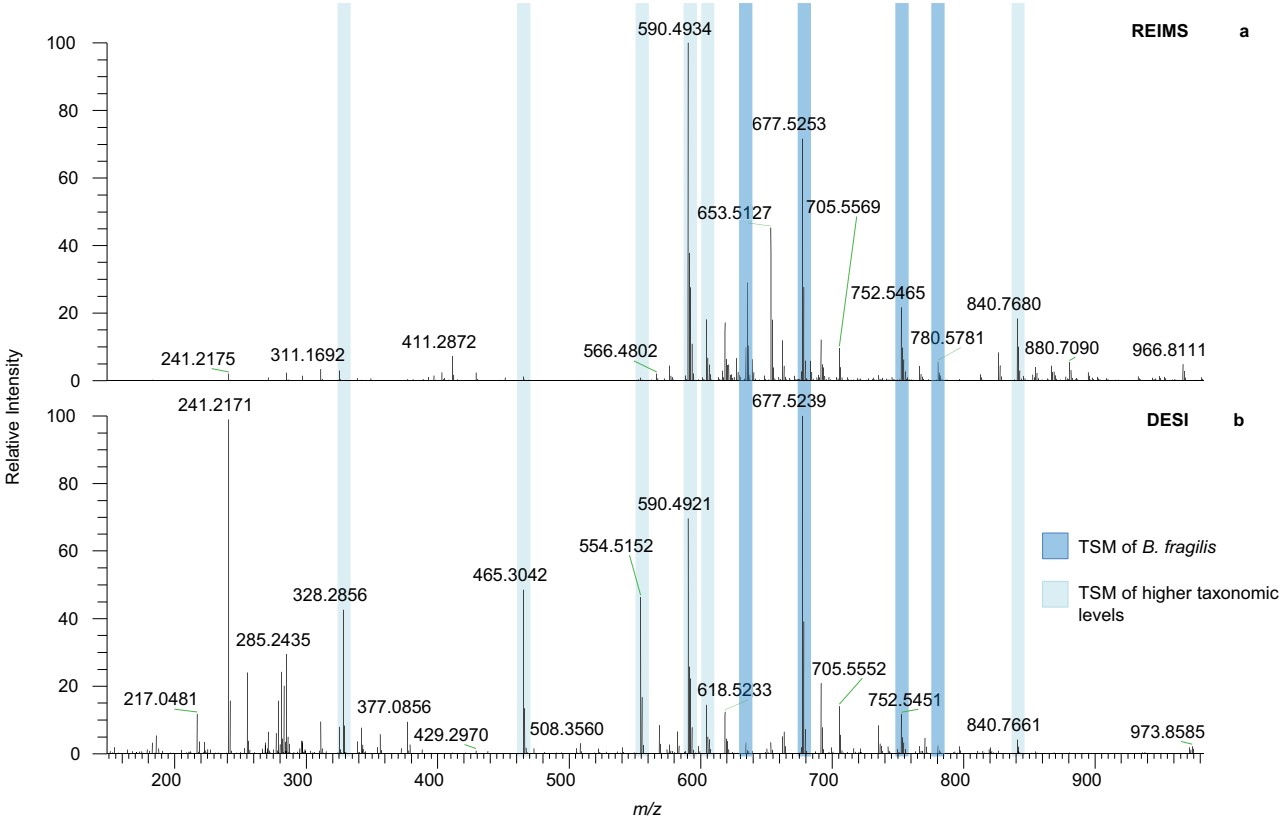

**Fig. 5 | Spectral comparison of *B. fragilis*. a** REIMS; **b** DESI-MS. TSMs are shaded in blue. Dark blue: *B. fragilis*-specific markers. Light blue: TSMs for higher taxonomic levels. Their *m/z* values and taxa are (from left to right): 328.286 (Bacteroidaceae, family), 465.304 (Bacteroidetes, class), 555.520 (Isotope of 554.5156, Bacteroidaceae, family), 590.493 (Bacteroidaceae, family, followed by isotopic TSM peaks at 591.497, 592.490, 593.493, 594.496), 604.509 (Bacteroidetes, class, followed by isotopic TSM peaks at 605.513 and 606.505), 634.521 (*B. fragilis*, species), 677.524 (*B. fragilis*, species, followed by isotopic TSM peaks at 678.529 and 679.530), 752.542 (*B. fragilis*, species), 781.582 (Isotope peak of 780.5734, *B. fragilis*, species), 840.768 (Bacteroidetes, class). **a, b** were averaged spectra of three technical replicates, collected on two different *B. fragilis* isolates. Relative peak intensities are shown as percentages compared to the base peak in each spectrum.

fragmentation data is available for TSMs, this comparison is based on exact masses only. The faecal microbiome is frequently used as a proxy for the gut microbiome[55], thus, the detected TSM are expected to show high similarity with those detected from tissues. 107 TSMs were detected in the LC-MS data (within a 10 ppm window), compared to 143 TSMs detected using DESI-MSI. The overlap between these two detected marker sets comprises 77 markers (see Venn diagram in Supplementary Fig. 6 and Supplementary Data 6). Most markers are thus detected using both instrumental modalities (54% and 72% for MSI and LC-MS data, respectively). A comparatively larger number of TSMs is detected in the MSI dataset, likely due to the dilution of local high bacterial load during the tissue homogenisation that is part of LC-MS workflows, a disadvantage that is avoided in imaging-based analysis. This is corroborated by bacteria being detected in hotspots of generally less than a square millimetre in size.

To assess whether the bacterial hotspots detected using TSMs correlate to sites of actual bacterial abundance, we have analysed tissues from a murine colorectal cancer model as well as a murine *H. pylori* gastric infection model using spatial metabolomics while a neighbouring section was subjected to 16S rRNA fluorescence in situ hybridisation (FISH) analysis. Such correlative imaging approaches are powerful in showing bacterial cell presence in tissues[14]. Figure 6a–d shows murine gastric tissue 1-week post-infection with *H. pylori*. The marker *m/z* 299.260 (Helicobacteriaceae family), possibly belonging to 11-MeO-heptanoic acid[56] was detected in a distinct hotspot pattern. 16S rRNA FISH analysis shows hotspots of bacterial presence at the same locations. As not many bacteria are capable of surviving in the harsh gastric environment, it is thus likely that these are indeed originating from *H. pylori*.

A more complex picture can be seen in murine colorectal cancer tissue shown in Fig. 6e, f. Complex microbial communities are expected in colorectal tissue[36,52]. Several markers belonging to the Bacteroidetes class, abundant intestinal microorganisms, are observed in this tissue, similar to those observed in human colorectal tissues (see Supplementary Figs. 3–5). TSMs, as well as 16S rRNA FISH imaging, reveal a pattern of microorganisms locating to hotspots. Discrepancies are expected, as the FISH probe coverage and TSM coverage are not identical. However, the pan-bacterial Eub338 probe should be detected in all positions where TSMs are detected. While allowing for some differences in localisation due to non-identical tissue sections, this seems to be the case (Fig. 6). Differences in the intensity of markers between FISH and TSMs might be related to actual local bacterial composition as well as differences in ionisation response between the different markers.

## Discussion

In order to address the challenges associated with detecting bacteria in complex clinical specimens using MS and assess the feasibility of the often-suggested concept of small molecular biomarkers, we compiled the most comprehensive database of bacterial small-metabolite profiles acquired using ambient, small-metabolite-based MS profiling techniques to date. Such a database has to adequately cover bacterial taxonomy at all levels and include a sufficient number of closely related bacterial species. As the database used here was acquired from blood culture isolates, it specifically reflects the bacterial complexity encountered in clinical microbiology settings. Using this database, we are able to assess the specificity of bacterial biomarkers in a realistic

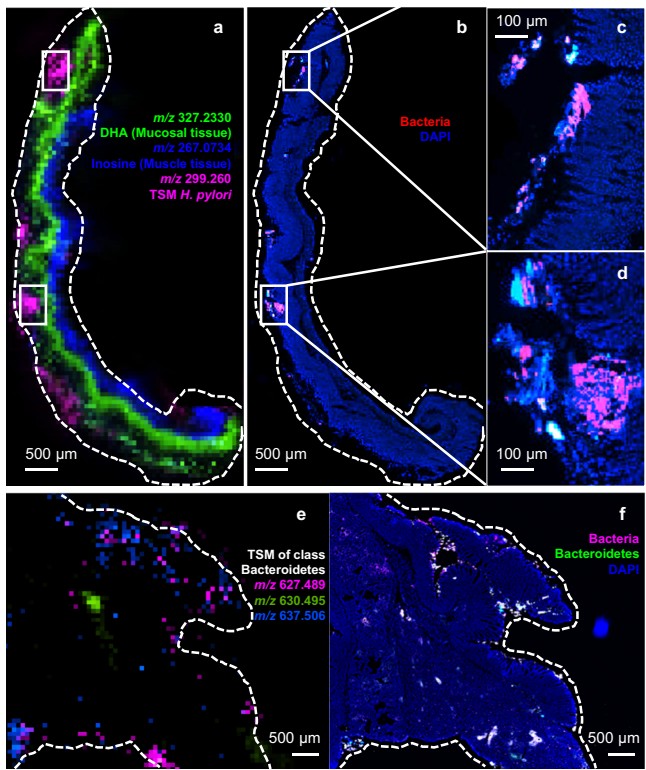

**Fig. 6 | Comparison between mass spectrometry images (MSI) and Fluorescence In Situ Hybridisation (FISH) images. a–d** Murine gastric tissue, one week after infection with *Helicobacter pylori*. **a** the overlayed DESI-MS image for two tissue markers and a TSM of *H. pylori* at *m/z* 299.260. MS spatial resolution: 75 µm. Scale bar: 500 µm. **b** the 16S rRNA FISH image of the neighbouring tissue section from **a**. Scale bar: 500 µm. Blue: DAPI probe for DNA. Red: Eub338 bacteria probe (appearing magenta through mixing with the co-localised blue DAPI probe). **c, d** Zoomed 16S rRNA FISH images, zoomed regions were boxed by white rectangles in **a** and **b**. Scale bar: 100 µm. **e, f** Tumour tissue of a mouse model for colorectal cancer (Apc^1638N/wt x pvillin-Kras^V12G). Scale bar: 500 µm. **e** Overlayed DESI-MS images for three different TSMs of class Bacteroidetes. MS spatial resolution: 75 µm. **f** 16S rRNA FISH image of the neighbouring tissue section to **e**. Blue: DAPI probe. Magenta: Eub338 bacteria probe. Green: CF319a probe for phylum Bacteroidetes, covering 90% of Flavobacteria, 90% Sphingobacteria and 30% Bacteroidetes[66,67]. Mass tolerance for MSI generation: 10 ppm. See Supplementary Fig. 7 for hematoxylin and eosin (H&E) staining results. The colour scales for MSI and FISH images represent the relative intensity of the signals (always increasing linearly from black to the marker colour). For FISH images, ×40 objective, the intensity threshold of individual colour channels was adjusted correspondingly to merge two/three colour channels and avoid overexposure. An additional example of a murine jejunum tissue can be found in Supplementary Fig. 8.

distinguish a certain species among a large cohort of other bacterial species) and sensitivity (*m/z* values can be detected directly in complex human matrices). We have chosen a univariate biomarker approach over multivariate approaches as this directly relates a single metabolic feature to a specific bacterial taxon. This approach facilitates detection in targeted metabolomic workflows, reduces bioinformatic requirements, and minimises spectral overlap with both the tissue matrix as well as commensal microorganisms.

We have found TSMs for ~30–40% of the clinically relevant bacterial groups that were eligible for testing (see Supplementary Fig. 2). While markers on high phylogenetic levels such as phylum were predominantly found for Bacteroidetes, markers on lower phylogenetic levels were detected for a wider group of bacteria, especially on family- and species-level. The number of TSMs found per taxon is an indication of the homogeneity of bacterial groups within each taxon and heterogeneity in comparison with other taxa. Especially among the phylum Bacteroidetes many unique spectral features were found, largely associated with sphingolipid metabolites. This is attributed to their special polysaccharide utilisation capacity as well as their long evolutionary process in the gut environment[57,58]. However, on the species-level, TSMs were found for 34 bacterial species, spanning over all investigated phyla.

To further promote the clinical translation of these markers, the identification of the molecular ID of each marker is a critical step in future TSM validation. Due to the high genetic and metabolic variability of microbial species, the molecular identity of many TSMs is currently unknown or only tentatively assigned and remains to be confirmed. Recently Zuffa et al. have developed a MS/MS-based microbial metabolite search platform termed MicrobeMASST[45], which allows users to search for known/unknown MS/MS profiles against public databases of microbial single cultures and map the results to taxonomic trees. Joint use with such platforms might further improve the identification and reliability of the TSMs. Knowing the metabolite in question will enable a more direct transfer of TSMs between different MS modalities and polarities through the tracking of different potential adducts. In this study, we have performed our analysis in negative ion mode only (adducts predominantly of the type [M-H]⁻ and [M+Cl]⁻ which can be easily distinguished based on isotopic patterns), and as a result, different adduct formation behavior was not taken into account. Due to the different chemical nature of the different TSMs, the exact ionisation behaviour and thus sensitivity in a clinical application for each marker will depend on its physicochemical properties and the matrix in question.

Expression of markers on higher phylogenetic levels (genus and above) was found to differ between the different members on that level, suggesting that TSMs to quantify bacteria is likely the most feasible on species-level. Marker abundance on species-level, however, was also found to be affected by different culturing conditions. TSMs will further need to be assessed whether they occur intracellularly or can be excreted or transferred using horizontal gene transfer. Thus, the current information output of the TSMs is similar to today's culturing-based clinical microbiology practice, indicating primarily the presence of certain bacterial species. If used in similar, tightly regulated biological matrices (such as blood or tissues), TSMs might be suitable for the quantification of bacteria; this, however, will need careful validation for each marker in each matrix to determine whether the required sensitivity and specificity can be achieved to positively identify and quantify a causative agent. These attributes will likely be maximised in a targeted approach and include additional dimensions such as collisional cross-section (CCS) values, fragmentation behaviour, and retention times.

Of those TSMs found, several interesting clinical applications are presenting themselves, such as in cases of *C. difficile*, *Fusobacterium nucleatum*, *Neisseria gonorrhoeae*, *P. aeruginosa*, *Streptococcus pneumonia,* and *Streptococcus pyogenes* or the *Helicobacteriaceae* family

setting for the first time. Compared to real bacterial complexity (over 1400 bacterial pathogens are described with a total number of bacterial species likely to be in the range of several millions)[34], the species coverage of our database is still low at 233 largely opportunistic pathogenic species. However, REIMS has been further developed into fully automated instrumentation comprising a colony picker and a laser for sampling and ionisation while simultaneously collecting stock for re-culturing or sequencing analysis[23,48]. Such a platform can be deployed for high-throughput, high-quality database acquisition to further increase coverage of microbial diversity. However, the complexity that will need to be considered is likely application-dependent: not all bacterial species are expected in all biological niches.

With this study, we particularly wanted to investigate whether the concept of bacterial small-metabolite biomarkers is feasible both in terms of specificity (*m/z* values are sufficiently distinctive to

(here only represented by *H. pylori*, resulting in a shift of all markers to family-level). *C. difficile,* for instance, is a common agent of nosocomial infections following antibiotic treatment, while *P. aeruginosa* is one of the main agents associated with flares of cystic fibrosis. *m/z* 286 is tentatively identified as 2-nonyl-3-hydroxy-4(1H)-quinolone (C9-PQS), a quorum-sensing molecule implicated in the virulence of *P. aeruginosa* strains, that is able to modulate the innate immune response of hosts during the *P. aeruginosa* infection progresses[59]. Bardin et al. have also identified this compound in *P. aeruginosa* isolates from cystic fibrosis patients using REIMS[21]. *N. gonorrhoeae* is a pathogen that is frequently screened for, as it causes the sexually transmitted genitourinary infection gonorrhoea, which can lead to disseminated disease and further negative outcomes such as septic arthritis and infertility if left untreated.

To assess whether TSM can be detected directly in complex tissue samples, we have screened several not-interdependent datasets from the gastrointestinal tract of both mice and humans for TSMs. TSMs were detected both in standard spatial metabolomics as well as bulk-based metabolomics workflows. The TSMs detected in either case were similar, and the bacteria detected were expected based on metagenomic profiling of these tissues (Supplementary Data 5 and Supplementary Fig. 6), comprising taxa such as Bacteroidetes, Clostridia, and Bifidobacteria.

In conclusion, TSMs can be useful tools for the rapid detection of bacteria in biofluids in first-tier screening applications. The ability to visualise bacteria in relation to metabolic niches, such as in cancer or infection scenarios, has the potential to advance our understanding of host-microbe interactions. Future studies are, however, necessary to carefully validate these markers in different biological and clinical scenarios. A distinguishing feature of the metabolome is that it can be influenced by outside factors in the environment, thus, interferences cannot be categorically excluded at this point, but potentially minimised through further TSM dimensions and targeted workflows. In practice, interferences both through matrices as well as other bacteria are potentially counterbalanced by a smaller number of bacterial species expected in a certain clinical or biological setting, determined by the conditions under which different bacteria can survive and grow. Identifying and taking only relevant species and matrices into account might further expand TSM numbers and specificity.

## Methods

### Ethical statement
The use of human colorectal tissue specimens and fecal samples was ethically approved by the institutional review board at Imperial College Healthcare NHS Trust under reference numbers 07/H0712/112 and SUR_JK_17_046, respectively. Written informed consent for sample collection was obtained from all patients. Animal experiments were carried out under institutional and national guidelines and regulations of Klinikum rechts der Isar, Technische Universität München. More detailed information on tissue collection and processing can be found in the Supplementary Methods.

### REIMS bacterial database acquisition.
All bacteria were isolated during routine practice at the NWLP clinical microbiology laboratory at Charing Cross Hospital, UK. Bacterial IDs were obtained using the Bruker Biotyper platform (identification through ribosomal protein profiles using MALDI-MS) by trained Biomedical Scientist microbiology staff during routine clinical microbiology practice following manufacturers protocols (expected misclassification rate 2–5%)[60]. For REIMS analysis, 0.5–1 mg of freshly cultured bacterial biomass was analysed using a bipolar sampling probe (bayonet-shaped irrigated bipolar forceps by Erbe Elektromedizin, Tübingen, Germany) directly connected to a Thermo Scientific Exactive mass spectrometer (Thermo Fisher Scientific Inc., Bremen, Germany) via a polymer tubing[25]. A power-controlled Valleylab Force EZc (Covidien, Dublin,

Ireland) was used as a radiofrequency alternating power supply at a setting of 60 W. The MS system was operated in negative ion mode covering a full scan mass range from *m/z* 150-2000 at a mass resolution of 50,000 (at *m/z* 200). MS instrument was calibrated using the commercially available Thermo Fisher Scientific PierceTM calibration mix before use with a home-built nanoelectrospray ionisation source. At least three technical replicates from different areas on the agar plate were recorded for each isolate; the criteria for inclusion of these replicates is defined in Pre-processing (below). Instrument parameters are given in Supplementary Table 1.

### Database composition.
Overall, the spectral database contains 3274 bacterial strains across 233 different bacterial species. As a training dataset to derive TSMs, a smaller database was created from this larger set that comprises spectral profiles of 597 strains, including every unique species and limiting the overall number of entries per species to a maximum of seven entries. These 233 species belong to 81 genera, 47 families, 23 orders, 11 classes, and 5 phyla of opportunistic pathogens and commensals. 22 groups were identified to genus-level only (as commonly indicated using an ending "sp.", as in "*Comamonas* sp."). As these were rare isolates, they were included in the database for better coverage of the clinically relevant bacterial pool. A representation of the dataset on phylum- to class- and order-level, respectively, is shown in Fig. 1. A detailed composition of this sample set can be found in Supplementary Data 1. Different culturing media (rich media only), culturing ages, and culturing atmospheres were included in the sample set where available. The full set of 3274 isolates can be found in MetaboLights MTLBS10328. The file names of isolates included in the training set are included in Supplementary Data 1.

### Pre-processing.
Profile mode spectra were converted from Thermo.raw format to.mz5 format using ProteoWizard msconvert tool (version 3.0.4043)[61] and imported into MATLAB v2023a (v9.14.0, MathWorks). For each of the 597 files, up to five individual scans from each file were picked if their total intensity was greater than three times of the mean of the total intensity in each file. The scans for each file were interpolated to a common *m/z* vector between 200-1000 and scaled according to total intensity. Peak picking was performed on each mean spectrum using MATLAB's *findpeaks* function (Signal Processing toolbox, v9.2) with a noise threshold set to include peaks with a threshold greater than twice the spectral intensity's 85th percentile; the median number of peaks detected across all files was 985. The frequencies of the peaks' *m/z* value were recorded across all files, and then smoothed in two stages; the first using a moving sum and the second with a moving Gaussian window. To account for increasing peak widths at higher *m/z* values, a 15 ppm window was used for each. Local maxima were identified in the smoothed spectrum, and those with a frequency of at least 10 (in the sum-smoothed frequency plot) were selected, which resulted in 5866 peaks.

### Univariate analysis.
The data matrix of size 597 × 5866 was normalised by total intensity. One-way analysis of variance (ANOVA) was calculated for each variable at each of the seven phylogenetic levels (Gram through to species), with the *p*-value recorded for each (two-tailed) test (Statistics and Machine Learning toolbox, v12.5). For any variance with $p < 0.05$, Tukey's Honestly Significant Difference (HSD) post-hoc test was performed. HSD calculates the confidence intervals for differences between the mean of two groups and if this interval spans zero then the difference between the two means is regarded as insignificant. The number of pair-wise differences for the group with the largest mean was determined and calculated for each of the seven phylogenetic levels. The emphasis has been placed on variables that are significantly higher in one group than in others, rather than variables that are absent in all but one taxonomical group. This approach enables finding markers that would have otherwise been excluded in

the likely scenario of groups containing misclassified bacterial species that do not show the same spectral feature. If, for any variable, the most different group at the species-level was not fully differentiated, (e.g., "*Comamonas* sp."), then it is not further considered.

**Comparisons and consistency.** As the univariate and post-hoc methods are performed independently, the groups with the largest numbers of differences may not be taxonomically compatible. Thus working upwards from the species level, each taxon is compared to its parent to determine if, for example, the genus is a member of the family. Where the parent taxon is empty (because $p > 0.05$), the consistency is assumed to be true. Taxons can only be considered specific as long as the phylogeny remains consistent from the Gram-level downwards; breaks in phylogenetic consistency prevent a variable from being considered specific at lower taxons.

**Receiver operating characteristic curve.** For each variable, and at each taxonomic level, a ROC curve was determined by comparing the most different group against all others (see Supplementary Fig. 9 for examples). The true and false positive rates (TPR and FPR) are determined, with specific features requiring a $TPR \geq 0.80$ and an $FPR \leq 0.10$ in the relevant phylogenetic level.

**Check for specificity.** For each variable, the specificity at each of the 7 taxons was determined using the following criteria: a) $p < 0.05$; b) true positive rate $\geq 0.8$; c) false positive rate $\leq 0.10$; d) at least 3 observations (e.g., bacterial profiles in the training set); e) at least one other similar group; f) significantly different to all other phyla/classes/orders, or all but 2 other families/genera/species; g) at species-level, not a "Genus sp.".

The all others-but-two metric for family, genus, and species was used for these levels to accommodate groups where the low number of observations makes it more challenging to differentiate higher-intensity groups that are statistically different (note that the selected group still has the largest mean of all groups). Thus for each variable, specificity (according to the criteria above) may be indicated at none, some or all taxons. Where any are specific, the lowest (most specific) phylogenetically consistent taxon is used. Of the 5866 variables, a total of 359 passed these requirements, and are listed in Supplementary Data 2.

**Multivariate methods.** Principal components analysis (PCA) was performed on total intensity normalised and $\log_{10}$ transformed data (added offset of 1).

**Validation of markers against pure bacterial cultures.** The detected features were assessed for their specificity using MS profiles of biologically independent bacterial isolates. The 'internal' validation set uses the remaining 2677/3274 isolates that were not included among the 597 isolates used for the determination of TSMs (see Supplementary Data 1). The external dataset consisted of 564 bacterial isolates spanning 38 species, analysed using a high-throughput REIMS platform as described in ref. 48. Agar plates with microbial colonies were placed into racks, and colonies of interest were manually selected via a visual interface on the Freedom EVO 75 robot (TECAN, Switzerland) equipped with a Pickolo platform (SciRobotics, Israel). A modified 200 μL pure tip stainless steel electrode probe (TECAN) was used to apply an electrical current (17 W, monopolar mode) via an ERBE IC 300 electrosurgical generator, rapidly heating the biomass to generate an aerosol. The aerosol was transported through a 1.5 m PTFE tube to a Xevo G2-XS Q-ToF mass spectrometer (Waters Corporation) with REIMS atmospheric pressure interface, operated in a negative ion mode of 50 to 2500 *m/z* range. Daily calibration with sodium formate ensured mass accuracy, and a 10 ng/μL leu-enkephalin solution was co-aspirated for internal lock mass calibration. Five measurements were

performed per isolate, with one colony used per measurement. A list of their taxonomy can be found in Supplementary Data 7. Scan detection in these files was performed as outlined above. The 359 TSM peaks were matched within ±15 ppm (internal) or ±50 ppm (external) and normalised to the total intensity over the 200–1000 *m/z* range. In each dataset, the area under the receiver operating characteristic curve (AUC ROC) was determined for each of the 359 variables where there were at least three observations in the relevant group. A total of 263 and 134 variables were able to be tested across the internal and external validation sets, respectively.

**Validation of markers in complex samples.** Different complex (pre-) clinical samples of different origins were tested for the detection of TSMs (see Supplementary Table 2). These include DESI-MS imaging datasets of the gastrointestinal tract of both human and murine origin (stomach, jejunum, colon, colorectal cancer) and LC-MS datasets of fecal specimens (Details on used samples can be found in Supplementary Methods). Only data in negative ion mode was used. All datasets have been collected independently from each other over several years on different instrumental platforms, in different facilities, and by different operators (see Supplementary Table 2).

**16S rRNA FISH analysis.** 16S rRNA FISH assays were carried out in sections cut at 10 μm from tissues embedded in hydroxypropyl methylcellulose and polyvinylpyrrolidone mix[62]. The sample preparation procedure can be found in the Supplementary Methods. The FISH probe Eub338 (5′-GCTGCCTCCCGTAGGAGT-3′, biomers.net GmbH, Ulm, Germany) containing a 594-label on the 5′ and 3′ prime end each was used as pan-bacterial marker. A CF319a (5′-TGGTCCGTGTCTCAGTAC-3′, biomers.net GmbH, Ulm, Germany) containing 4×488-label (on the 5′ and 3′ prime end each and two in the middle) was applied to target the phylum Bacteroidetes. The CF319a probe covers 90% Flavobacteria, 90% Sphingobacteria, and 30% Bacteroidetes[63]. For the double hybridisation, each probe solution (50 ng/μL) was mixed with hybridisation buffer (35% formamide, 900 mM NaCl, 20 mM Tris-HCl (pH 8.0), 10% dextran sulphate (*w/v*), 0.02% SDS, 1% blocking reagent (Roche, Basel, Switzerland)) at a ratio of 1:13 (*v/v*). Probe hybridisation was performed at 46 °C in the dark under a saturated formamide-water atmosphere for 4 h. Cell nuclei were stained with 4′,6-diamidino-2-phenylindole (DAPI) for 10 min in the dark at room temperature. Before scanning, slides were mounted with a mix of Vectashield® antifade mounting medium and Citifluor™ mounting medium (1:3 *v/v*). Fluorescence overview images of the whole tissue section were taken by an Olympus BX53 compound microscope (Olympus, Tokyo, Japan) equipped with an ORCA Flash 4.0 (Hamamatsu Photonics K.K, Hamamatsu, Japan) camera and ×40 objective, using imaging software (Olympus cellSens Dimension v2.3). Fuji ImageJ (v1.54 f)[64] software was used to generate the figures. The three channels were merged into one, and the intensity threshold of each channel was linearly adjusted separately to avoid overexposure. Raw image results can be found at Figshare (https://figshare.com/articles/figure/_/26504344).

**Statistics and reproducibility.** Sample sizes for the TSM training and test databases were determined by the availability of specimens and bacterial strains from clinical sources. Data was recorded in a randomised fashion over several instances. No data were excluded from the analyses, except in cases where mass spectra did not meet the established quality control thresholds during pre-processing (e.g., low signal intensity or significant noise).

For the assessment of the quantitative nature of TSMs, 11 independent biological isolates were cultured on 5 different culturing media and subjected to analysis using REIMS. One file was excluded as it was corrupted. Bacteroides data was taken from the training database. Details of quality control and spectral pre-processing are

provided in the Methods section. The investigators were not blinded during data acquisition. Statistics were performed using automated algorithms to minimise human bias.

The reproducibility of results was assessed through multiple validation approaches: (i) Internal Validation: Independent bacterial isolates ($n = 2677$) excluded from the training database were analysed. (ii) External Validation: An independent dataset of bacterial isolates ($n = 564$) was analysed on a different instrument, and the results were cross-validated. (iii) Cross-method Validation: TSM detectability was confirmed in 44 human colorectal tissue samples using DESI-MSI and 48 human faecal samples using LC-MS, further DESI-MSI datasets were recorded in a different laboratory (*H. pylori*-infected murine gastric tissue, murine jejunum, murine olorectal tumour), see Supplementary Data 4. iv) FISH Validation: 16S rRNA FISH was performed on tissue sections to validate the presence and localisation of bacteria. For FISH analyses, two slides for nine murine gastric tissues post-infection with *H. pylori*, one slide for one healthy colon tissue (wild-type BALB mouse), one tumour tissue for colorectal cancer (Apc 1638 N/wt × pvillin-Kras V12G mouse), and two jejunum tissues (wild-type BALB and Apc 1638 N/wt × pvillin-Kras V12G mice) were analysed. Similar results were obtained across all tissues, with bacteria detected where TSMs were detected using MSI.

### Reporting summary

Further information on research design is available in the Nature Portfolio Reporting Summary linked to this article.

## Data availability

The bacterial mass spectrometry data generated in this study have been deposited in the MetaboLights database under accession code MTBLS10328. The DESI mass spectrometry imaging data of human colorectal tissues data used in this study have been deposited in the MetaboLights database under accession code MTBLS289. The DESI mass spectrometry imaging data of murine tissues generated in this study have been deposited in the MetaboLights database under accession code MTBLS10846. The mass spectrometry imaging data used in this study are also available in the METASPACE database [access link supplied in Supplementary Data 4]. The FISH data are available at Figshare [https://figshare.com/articles/figure/_/26504344]. The liquid chromatography-mass spectrometry (LC-MS) data for human faecal samples and the external bacterial mass spectrometry (REIMS) data used in this study are deposited (yet currently still in curation) under MetaboLights identifiers MTBLS11775 and MTBLS11776, respectively. Until publication, data can be requested from Dr. James McKenzie (j.mckenzie@imperial.ac.uk). Source data are provided in this paper.

## Code availability

The code for TSM determination is available via the public GitHub repository https://github.com/jsmckenzie/bacterialTSM (https://doi.org/10.5281/zenodo.14169354)[65], which also includes worked examples for a subset of variables.

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

## Acknowledgements

N.S., W.C., and M.Q. acknowledge funding from the Technical University of Munich and the Deutsche Forschungsgemeinschaft (DFG, project no. 505372148). M.Q. acknowledges funding from the Chinese Scholarship

Council (CSC, no. 202208320033). P.P., J.K., Z.T., and J.M. acknowledge funding from Cancer Research UK (C52720/A25038) and the NIHR Imperial BRC (Molecular Phenomics). We thank the National Phenome Centre at Imperial College London for assistance with LC-MS/MS data acquisition. K.J. acknowledges funding from the Deutsche Forschungsgemeinschaft (DFG, project no. 395357507—SFB 1371). MS and ML acknowledge financial support from the DFG within the Collaborative Research Center (CRC) 1182 on the Origin and Function of Metaorganisms (Project-ID 261376515) and the Max Planck Society.

## Author contributions

W.C., M.Q., L.Z., T.R., and N.S. performed DESI-MS and REIMS experiments. P.P. performed LC-MS experiments. J.M., N.S., W.C., M.Q. performed data analysis and interpretation. J.M. performed algorithm development. W.C., M.Q., N.S. performed metabolite identification. N.S., J.M., and Z.T. conceptualised this study. M.S., M.L. performed the FISH analysis. J.K. performed colorectal study sample acquisition and organised ethical procedures. M.R. supplied clinical microbiology isolates. K.J. supplied murine colon and jejunum tissues. RM supplied *H. pylori*-infected murine stomach tissue. RG performed histopathological examinations. W.C., N.S., and J.M. wrote the first draft, and all authors read and revised the final draft.

## Funding

## Competing interests

The authors declare no competing interest related to this work.
