## [Peer Review File · Nature Communications]

Universal, untargeted detection of bacteria in tissues using metabolomics workflows

Corresponding Author: Professor Nicole Strittmatter

Version 0:

Reviewer comments:

Reviewer #1

(Remarks to the Author)
Review for Nature Communications

Manuscript Number: NCOMMS-24-09311

Title: Universal, untargeted detection of bacteria in human tissues using spatial metabolomics

Authors: Wei Chen, et al.

Review:

The authors describe a biomarker-based, hierarchical strategy that utilizes conserved bacterial small molecular metabolites and lipids for direct mass spectrometric detection of bacteria in clinical samples. They created a metabolic library of 232 bacterial species, then mined the library for markers showing specificity at various taxonomic levels. They found 359 bacterial taxon-specific markers (TSMs) via a combination of univariate and multivariate statistical analysis methods. They then applied the TSM markers to detect in situ bacteria directly from complex clinical specimens, using healthy and cancerous colorectal tissues as well as fecal samples. TSMs were found in >90% of samples tested, suggesting the general applicability of this methodology to detect bacterial presence with standard MS-based analytical methods.

The authors identified 213 bacteria, shown in Table S2.

More specifically, the authors used both REIMS and DESI-MS for bacterial small molecule detection to identify TSMs. To demonstrate the applicability of these taxonomical markers for the detection of bacteria in complex biological, human-derived samples, the authors visualized the presence and distribution of bacteria in 44 human colorectal tissues recorded using DESI-MSI. Liquid chromatography MS was used on fecal samples to detect TSMs and specific bacteria.

Overall, the authors use a relatively novel approach to identify TSMs and to visualize specific bacteria in human tissue and identify bacteria in human fecal samples. This is an important step towards identifying bacteria in tissues. This involved a tremendous amount of work to define TSMs and then to prove the specific bacteria could be identified in human tissue. This area of research needs to be encouraged as it has significant promise for the future.

Concerns:

1) There is internal but not external validation to define TSMs. This would need to be validated in a separate collection of samples. However, the defined TSMs were used to visualize the presence and distribution of bacteria in 44 human colorectal tissues recorded using DESI-MSI. In addition, to show that the TSM markers can also be detected using LC-MS datasets and as a further step of validation of these markers, the authors screened a set of 48 fecal samples for TSMs. Overall, for DESI-MSI, 143 markers were detected, while 107 markers were detected for LC-MS data. 77 markers were shared by both techniques. Thus, the majority of TSM markers were detected using both instrumental modalities (54% and 72% for MSI and LC-MS data, respectively). This would also need to be validated in another facility to see if this methodology is generalizable and is specific. Thus validation studies are highly encouraged to show validity of the

methodology.

2) Bacterial quantification is a problem using the metabolomics methodology. The number of TSMs found cannot be directly compared to the copy-numbers found per bacterial taxon using molecular methods as the number of TSMs is not linearly related to the number of bacterial cells. Thus, the predominant infective agent may be difficult to ascertain with specificity using this methodology. In the real world, clinicians need to know what bacteria is the most important for the infected patient to be treated. How would the authors propose getting around this problem? This needs to be discussed in the discussion.

3) It would be useful to further explain feature selection rules for the TSM identification.

4) The authors need to expand the potential limitations to this research.

5) There are a number of typos that need to be addressed.

Reviewer #2

(Remarks to the Author)

The authors present their work on untargeted detection of bacteria in human tissues using spatial metabolomics.

This is a very important area and relevant to Nature Communications. I suggest to accept it after a minor revision.

The authors did a tremendous work collecting the data. Data analysis is also appropriate. However, I believe the presentation of the data and results are a bit foggy. The authors should clearly and simply summarize their findings. Can or cannot their method be used? What needs to be done to have this method a fully deployable approach in the clinical field?

More on the experimental side: it was not clear for the reviewer if the training and validation data were acquired from different biological replicates or not? If not, the results can be seriously biased.

The DESI data should go into the supplemental, not directly related to the subject in hand.

The text a bit dense, took a lot of effort to go through it. If the authors could move some less relevant part into the supplemental, that would be great.

PS: Please fix the numerous typos in the manuscript.
were used in the original training dataset.

Reviewer #3

(Remarks to the Author)

The work presents a potentially useful approach to detecting bacterial species directly in clinical samples/tumors. This has potential use in several domains of biomedicine. The analysis of the number of tumor samples for method application is a nice application of the proposed methods. However, using multiple MS technologies (REIMS, DESI, and MALDI) for feature comparison and as training datasets to identify TSMs, needs further validation. In the current form, it is not convincing that the biomarkers reported will be widely applicable to detect bacterial species in other clinical samples. In case the author wants to propose the workflow, I would suggest having more details on methodological validation and making computational tools/code publicly available for other researchers to reproduce the workflow.

More detailed comments can be found below:

Introduction: The author highlighted four challenges for the wide applicability of such methods in clinical samples. How this study addressed all four or any of those challenges is not clearly explained or well understood by the readers.

Abstract: Typo error at "Samsamples."

Results: The data shown in Figures 1 to 3 are generated with REIMS, which is a different ionization technique than MALDI and DESI. Have the authors considered this aspect with direct comparison of features or transfer of TSMs?

Figure 1: It's hard to read the labels in the Sunburst plot. More description in legends is needed for the general scientific audience to understand the message of these plots. The interpretation of the plots is just to show the database composition without rationale or an overview of the selected database? Why were m/z peaks from 50 to 200 Da excluded during analysis or not acquired?

"5866 unique m/z values were detected over the entire training dataset." This number of unique m/z is too large. Does it mean in each spectral data file? Out of how many m/z values in total?

"359 features were found to be specific at a certain phylogenetic level and are considered TSMs." It sounds like each TSM is with 359 features exactly associated with each phylogenetic level? Or are they totally unique features?

Figure 2: It mentions "phylum of each of the 597 observations," which are again features. How do they arrive? Earlier, 359 features were considered to be TSMs.

Figure 2A) The scores for PC1 and PC2 are around 6%, which is a significantly low representation of variance across datasets.

"Of these, a total of 359 features were found to be specific at a certain phylogenetic level and are considered TSMs": Are these 359 TSMs not mentioned or described in Figure 2?

B) What does the TSM across m/z distribution show? The AUC specificity indicated by circle size seems relatively weak.

Since the majority (>80%) of TSM markers are found for the members of the phylum Bacteroidetes, does it skew the further data analysis towards a particular phylum or species?

"Of the 116 species-specific TSMs, 35 are identified as isotope peaks," which means 30% of total peaks are isotopic. I am not sure if this will not affect the downstream predictions and data analysis.

Lipid identifications need more confirmation at Level-1 whenever possible to improve the reliability of reported lipid biomarkers:

-Just using exact mass search on such datasets will not give reliable IDs or even tentative IDs. Consider using MS/MS data for IDs.

-Please describe how m/z 604.509 is identified as ceramide 35:0;O3, as exact mass or adduct masses are not matching. What does this statement mean? Is all datasets cross-verified for IDs with MALDI-MS for REIMS datasets? "All files used in this study were identified with MALDI-MS (using a Bruker Biotyper)."

- Similar descriptions for the ID of m/z 949.777 are required. Is there any reference or database citation available for the identification of "C15:0 substituted phosphoglycerol dihydroceramide"?

- There is only one (ref. 47) mentioned for these identifications in *Porphyromonas gingivalis*. However, the reference does not report exact masses or any other standard identification procedures.

- The author is missing a critical metabolite identification process here without any justification: "As for most TSMs currently, no fragmentation data is available for higher-level identification and comparisons for this proof of principle are made based on exact mass only."

- It is fairly straightforward to perform MS/MS with standard LC-MS analysis and on-tissue MS/MS with MALDI and DESI. The justification in the Discussion section is not convincing. "The presented approach was bioinformatically adjusted to allow the occasional misidentification."

Is the data submitted mostly acquired in negative mode? Is this the case with all other datasets?

Validation using independent data:

How can splitting the same data be qualified as "independent data," as mentioned in "Of the 3274 files forming part of the bacterial database, the remaining 2677 files not included in the training set were used as a test set for validation"? The independent datasets should be acquired or originated independently. What were the criteria for this split of data files?

Overall, the validation part is weak and needs more description and data analysis.

Feasibility of detecting TSM in colorectal cancer specimens:

44 tumors were analysed with DESI-MS, which is again a different ionization method. How are the previous datasets, TSMs, and models derived from different platforms transferable?

Figure 6: Particular m/z from TSMs were associated with bacteria in tumors. Other than the H&E image showing necrotic area, there should be other imaging such as IHC or bacteria-specific Immunofluorescence stained images for cross-validation.

Why was LC-MS analysis not performed on tumor samples itself for confirmation of identification? What is the link between 48 fecal samples and tumor samples or training datasets?

Version 1:

Reviewer comments:

Reviewer #1

(Remarks to the Author)

Review of revised Manuscript # NCOMMS-24-09311A

Title: Universal, untargeted detection of bacteria in human tissues using spatial metabolomics

Authors: Wei Chen, et al.

Review

This is a revision of a previously submitted manuscript. Again, this reflects a great deal of work in an area that does hold promise but definitive validated results are important for the future.

Concerns:

1) The authors have answered this reviewer's questions, but for example, on bacterial quantification, the answer is not direct and does meander. The answer is that it is not possible at this time but with more experience and more validation of markers it may be possible in the future. Because of this approach, this reviewer is not convince the manuscript is better. By way of answering the questions the manuscript appears more disjoint and possibly harder for the uninitiated reader to understand. Most readers do not know a lot about metabolomics and this needs to be considered when writing the manuscript. This is mostly stylistic.

2) Where is figure 4 in the manuscript. It goes from Fig. 3 To Fig. 5.

3) Line 275. The sentence is not complete.

4) Line 283. What does the label (A15 10cm S3) represent? Please explain.

(Remarks on code availability)

Reviewer #3

(Remarks to the Author)

The authors have considerable efforts in revising the manuscript by including additional validation, comparison of FISH and MSI images and explaining metabolite identification and confidence levels.

It will be useful to include some more details (which authors explained in ruttal such as FISH and MSI comparison, data analysis methods and lipid identification interpretation) directly into the manuscript or supplementary information. These details may not be relevant to main conclusions but can be valuable to working researchers in the field.

(Remarks on code availability)

Legend: The original comments made by the reviewers have been sequentially numbered for better reference. They are in navy blue font. Our responses are in black, while direct excerpts from the manuscript are in font Times New Roman.

Reviewer #1 (Remarks to the Author):

1) There is **internal but not external** validation to define TSMs. This would need to be validated in a separate collection of samples. However, the defined TSMs were used to visualize the presence and distribution of bacteria in 44 human colorectal tissues recorded using DESI-MSI. In addition, to show that the TSM markers can also be detected using LC-MS datasets and as a further step of validation of these markers, the authors screened a set of 48 fecal samples for TSMs. This would also need to be **validated in another facility** to see if this methodology is generalizable and is specific. Thus validation studies are highly encouraged to show validity of the methodology.

More validation studies have been included in response to this comment as well as comments 6, 31, 33 and 34. These comprise validation on external pure bacterial cultures and independent data recorded on murine colon and stomach specimens and cross-examined using FISH. The primary scope of this study was to test whether bacterial-specific metabolic markers are a viable proposition in general. This approach is frequently suggested,¹⁻³ however, has never been tested in a dataset of sufficient bacterial complexity. We have been pleased to find that while this approach is not applicable to every individual bacterial species, it does seem applicable to what we found to be 30-40% of bacterial species tested in this study. Comparing bacterial profiles of 232 species (this is an order of magnitude higher in species numbers than previously used sample sets) against each other, we were able to detect 359 specific markers. We have validated these markers against bacterial profiles obtained on an independent instrumental platform to test whether they have sufficient specificity to identify pure bacterial cultures. This data is now included in the validation section and summarised in Figure 4 (also see comment 7).

Whether any of these TSMs have sufficient specificity to be used in for instance a clinical test will depend heavily on the biological matrix the bacterium should be detected in. The LC-MS data of faeces used in our initially submitted manuscript version was recorded independently, in an independent core facility (the National Phenome Centre, Imperial College London) on a separate campus, by independent personnel on a different instrumental platform. This has been made more clear in the manuscript. We have further in the revision of this manuscript analysed several tissues of murine origin that were recorded in our facilities in Munich (see Figure 6 and Supplementary Fig. 8).

It is however within the scope of this study not feasible for us to fully validate the specificity of 359 TSMs in all possible tissues and biofluids. Thus, we have added a section to the discussion detailing the need for meticulous validation of each marker in the matrix of choice. Thus, the primary route to translation will likely be via targeted methods. Here, additional parameters such as CCS values, fragmentation behaviour or retention time might introduce additional specificity for a targeted application.

“With this study, we particularly wanted to investigate whether the concept of bacterial small metabolite biomarkers is feasible both in terms of specificity (m/z values are sufficiently distinctive to distinguish a certain species among a large cohort of other bacterial species) and sensitivity (m/z values can be detected directly in complex human matrices). [...] We have found TSMs for approximately 30-40% of the clinically relevant bacterial groups that were eligible for testing (see Supplementary Fig. 2). While markers on high phylogenetic levels such as phylum were predominantly found for *Bacteroidetes*, a variety of markers were identified for lower phylogenetic levels for a wider group of bacteria, especially on family- and species-level.”

“This results in the need to ultimately perform careful validation for each marker with regards to specificity and interference in the required host matrices to determine whether the required sensitivity can be achieved to positively identify and quantify a causative agent. These attributes will likely be maximised in a targeted approach and including additional dimensions such as collisional cross-section values, fragmentation behaviour and retention times.”

2) Bacterial quantification is a problem using the metabolomics methodology. The number of TSMs found cannot be directly compared to the copy-numbers found per bacterial taxon using molecular methods as the number of TSMs is not linearly related to the number of bacterial cells. Thus, the predominant infective agent may be difficult to ascertain with specificity using this methodology. In the real world, clinicians need to know what bacteria is the most important for the infected patient to be treated. How would the authors propose getting around this problem? This needs to be discussed in the discussion.

In principle, if work is performed within the linear range of the instrument, each TSM marker can be quantified. When working under similar conditions (matrices) for the same bacteria, this might translate to the ability to quantify the bacterium, especially when using species-level TSMs. This however needs to be carefully assessed in future, more application-directed studies. Here, we have observed markers to change in principle both among different culturing conditions and different species of the same higher taxon (see Figure 3). A new subheader (“Ability of TSMs to quantify bacteria”) was added to the results section to discuss the quantitative nature (or lack thereof) of the TSMs in more detail.

Clinicians are unfortunately faced with a similar problem today regarding a lack of quantitative information. In the culturing-based approach used in most clinical microbiology laboratories, there is equally no knowledge available on the bacterial quantities. Thus, our approach is rather similar to current microbiology practice, delivering mainly knowledge on who is there, while a clinical practitioner has to decide which bacterium is the causative agent. However, the timescale with which this information can potentially be delivered is shorter in an MS-based approach than a culturing-based approach. We do believe, that any information supplied on a short time scale will aid in making an educated guess for the first line of treatment before a causative agent is formally identified. For the short-to mid-term, the requirement of a clinical microbiologist to assign the infectious agent will remain based on the bacteria detected.

While the clinical application is of course apparent, we refrain at this point from the attempt to validate any specific one in particular. Our aim was primarily to establish whether a biomarker approach is feasible at all, both in terms of specificity and sensitivity. For this purpose, we have analysed data sets that were taken from independent MS-based studies where the primary goal was not the detection of small molecules of bacterial origin.

We have expanded the discussion on the requirement to carefully validate each marker regarding sensitivity and specificity. We have also included a section to position our technique and its route into widespread use. Please further refer to responses to comments 4, 6 and 12.

“TSMs on higher phylogenetic levels are likely unsuitable to quantify bacteria directly. Intracellular TSMs on species-level, if used under similar conditions (biological matrices) might be suitable for quantification of bacteria, this however will need careful validation for each species in each matrix. This current non-quantitative nature of the TSMs makes the information output similar to today’s culturing-based clinical microbiology practice. [...] Due to the different chemical nature of the different TSMs, the exact ionisation behaviour and thus sensitivity in a clinical application for each marker will depend on its physicochemical properties and the matrix in question.

It will be further imperative to study whether TSMs occur intracellularly or can be excreted and whether they can be transferred using horizontal gene transfer. This results in the need to ultimately perform careful validation for each marker with regard to specificity and interference in the required host matrices to determine whether the required sensitivity can be achieved to positively identify and quantify a causative agent. These attributes will likely be maximised in a targeted approach and including additional dimensions such as collisional cross-section values, fragmentation behaviour and retention times. [...]

In conclusion, we believe that taxon-specific markers can be useful tools for rapid detection of bacteria in biofluids in first-tier screening applications. The ability to visualise bacteria in relation to metabolic niches, such as in cancer or infection scenarios, has the potential to advance our understanding of host-microbe interactions. Future studies are however necessary to carefully validate these markers in different biological and clinical scenarios. A distinguishing feature of the metabolome is that it can be influenced by outside factors in the environment, thus, interferences cannot be categorically excluded at this point, but potentially minimised through further TSM dimensions and targeted workflows. In practice, interferences both through matrices as well as other bacteria are potentially counterbalanced by a smaller number of bacterial species expected in a certain clinical or biological setting, determined by the conditions under which different bacteria can survive and grow. Identifying and taking only relevant species and matrices into account might further expand TSM numbers and specificity.

3) It would be useful to further explain feature selection rules for the TSM identification.

We have added the following explanations to the Methods section:

“Comparisons and consistency. As the univariate and post-hoc methods are performed independently, the groups with the largest numbers of differences may not be taxonomically compatible. Thus working upwards from the species level, each taxon is compared to its parent to determine if, for example, the genus is a member of the family. Where the parent taxon is empty (because $p > 0.05$), the consistency is assumed to be true. Taxons can only be considered specific as long as the phylogeny remains consistent from Gram-level downwards; breaks in phylogenetic consistency prevent a variable from being considered specific at lower taxons.

Receiver operating characteristic (ROC) curve. For each variable, and at each taxonomic level a ROC curve was determined, by comparing the most different group against all others (see Supplementary Fig. 9 for examples). The true and false positive rates (TPR and FPR) are determined, with specific features requiring a $TPR \geq 0.80$ and an $FPR \leq 0.10$ in the relevant phylogenetic level.

Check for specificity. For each variable, the specificity at each of the 7 taxons was determined using the following criteria: a) $p < 0.05$; b) true positive rate ≥ 0.8 ; c) false positive rate ≤ 0.10 ; d) at least 3 observations (e.g. bacterial profiles in the training set); e) at least one other similar group; f) significantly different to all other phyla/classes/orders, or all but 2 other families/genera/species; g) at species level, not a “Genus sp.”.

The all-others-but-two metric for family, genus and species was used for these levels to accommodate groups where the low number of observations makes it more challenging to differentiate higher-intensity groups that are statistically different (note that the selected group still has the largest mean of all groups). Thus for each variable, specificity (according to the criteria above) may be indicated at none, some or all taxons. Where any are specific, the lowest (most specific) phylogenetically consistent taxon is used. Of the 5866 variables, a total of 359 passed these requirements, and are listed in Supplementary Data 2.”

4) The authors need to expand the potential limitations to this research.

This comment shows similarities to comment 2, 6 and 12. We have restructured and expanded on these aspects in the results and discussion section and hope that the current version is adequate.

We want to specifically point out the following new passages in the discussion section:

“Compared to real bacterial complexity (over 1400 bacterial pathogens are described with a total number of bacterial species likely to be in the range of several millions),³⁴ the species coverage of our database is still low at 232 largely pathogenic species.

[...] Due to the high genetic variability of microbial species as compared to more well-studied mammals, the molecular identity of many TSMs is currently unknown or only tentatively assigned and remains to be confirmed.

[...] This current non-quantitative nature of the TSMs makes the information output similar to today’s culturing-based clinical microbiology practice.

[...] It will be further imperative to study whether TSMs occur intracellularly or can be excreted and whether they can be transferred using horizontal gene transfer. This results in the need to ultimately perform careful validation for each marker with regards to specificity and interference in a variety of different host matrices to determine whether the required sensitivity can be achieved to positively identify and quantify a causative agent.”

5) There are a number of typos that need to be addressed.

We apologise for our earlier oversight. We have carefully proofread the manuscript and hope this aspect has been improved.

Reviewer #2 (Remarks to the Author):

6) The authors did a tremendous work collecting the data. Data analysis is also appropriate. However, I believe the presentation of the data and results are a bit foggy. The authors should clearly and simply summarize their findings. Can or cannot their method be used? What needs to be done to have this method a fully deployable approach in the clinical field?

We have restructured both results and discussion in order to address this point and related points (comments 2, 4, 12). In response to this comment we want to particularly point out the following passages:

“[...] We have found TSMs for approximately 30-40% of the clinically relevant bacterial groups that were eligible for testing (see Supplementary Fig. 2). While markers on high phylogenetic levels such as phylum were predominantly found for Bacteroidetes, a variety of markers were identified for lower phylogenetic levels for a wider group of bacteria, especially on family- and species-level.”

“In conclusion, we believe that taxon-specific markers can be useful tools for rapid detection of bacteria in biofluids in first-tier screening applications. The ability to visualise bacteria in relation to metabolic niches, such as in cancer or infection scenarios, has the potential to advance our understanding of host-microbe interactions. Future studies are however necessary to carefully validate these markers in different biological and clinical scenarios. A distinguishing feature of the metabolome is that it can be influenced by outside factors in the environment, thus, interferences cannot be categorically excluded at this point, but potentially minimised through further TSM dimensions and targeted workflows. In practice, interferences both through matrices as well as other bacteria are potentially counterbalanced by a smaller number of bacterial species expected in a certain clinical or biological setting, determined by the conditions under which different bacteria can survive and grow. Identifying and taking only relevant species and matrices into account might further expand TSM numbers and specificity.”

7) More on the experimental side: it was not clear for the reviewer if the training and validation data were acquired from different biological replicates or not? If not, the results can be seriously biased.

Yes, these were biologically independent biological replicates, isolated from separate patients. They have been recorded on the same instrumental platform over the course of three years. This set is in the new manuscript version considered as “internal” validation.

We have additionally now also performed validation on a fully separate cohort of pure bacterial cultures (15 isolates each from 38 species) that was recorded by different operators on a different instrumental platform in which ablation and ionisation were performed using a laser (LDI-REIMS) rather than Joule heating (conventional REIMS) deployed in the original database files and detection by a TOF rather than an Orbitrap mass analyser. This dataset is considered as “external” validation. The amended methods and results sections and the resulting Figure are shown below. Please also refer to comment 31.

Materials and Methods section:

“Validation of markers against pure bacterial cultures. The detected features were assessed for their specificity using MS profiles of biologically independent bacterial isolates. The ‘internal’ validation set uses the remaining 2677/3274 isolates that were not included among the 597 isolates used for the determination of TSMs (see Supplementary Data 1). The ‘external’ dataset (collected using a Waters Xevo G2-XS as described in reference ⁵⁹ consists of 564 bacterial isolates spanning 38 species. A list of their taxonomy can be found in Supplementary Data 7. Scan detection in these files was performed as outlined above. The 359 TSM peaks were matched within ± 15 ppm (internal) or ± 50 ppm (external) and normalised to the total intensity over the 200-1000 m/z range. In each dataset, the area under the receiver operating characteristic curve (AUC ROC) was determined for each of the 359 variables where there were at least three observations in the relevant group. A total of 263 and 133 variables were able to be tested across the internal and external validation sets, respectively.”

Results section:

“**Validation of biomarkers on pure bacterial cultures** Testing whether TSMs can be used to identify pure bacterial cultures was performed using two datasets. The ‘internal’ set consists of the 2677 bacterial profiles from bacterial isolates that were excluded from the training set (n=597 profiles) used to derive TSMs. The ‘external’ set contains 564 spectral profiles recorded on another independent cohort of clinical isolates on an Xevo G2-XS equipped with a REIMS atmospheric pressure interface.⁴⁸ Peak intensities for the 359 variables were extracted from all files, using the same methodology as for the training database. As a consequence of the compositions of the validation sets, not all TSMs were tested due to having fewer than 3 bacterial profiles in the dataset. In the internal dataset AUCs for 263 variables were determined, while for the external dataset this number is 133. These results are summarised in Figure 4 as both boxplots and histograms. The median (interquartile range, IQR) AUC values for internal and external variables are 0.997 (0.047) and 0.985 (0.070), respectively. The results show that many of these features are highly specific when using the same

instrumental platform (unmodified Thermo Exactive classic), and moderately lower when using an entirely different instrumental setup.”

Figure 1. Validation of TSMs using two validation datasets. AUCs were determined for each variable where there were at least 3 bacterial profiles/observations in each dataset. *a* Boxplots showing AUC for each taxon. No order-level TSMs could be tested in the external dataset and only one at the genus-level. *b* and *c* Histograms of AUC from 0.4-1.0 in bins of 0.05 for internal (*b*) and external (*c*) validation sets, respectively. The median values (and inter-quartile ranges) for internal and external markers are 0.997 (IQR 0.047) and 0.985 (IQR 0.070). IQR: interquartile range.

8) The DESI data should go into the supplemental, not directly related to the subject in hand.

The mentioned section on detection of TSMs was largely moved to the Supplementary Information as requested. As Validation was an issue raised by several of the reviewers, we have instead expanded on the validation paragraph in a more general fashion, of which the DESI data mentioned now makes a much smaller part.

9) The text a bit dense, took a lot of effort to go through it. If the authors could move some less relevant part into the supplemental, that would be great.

The following passages were moved into the SI in response to this and comment 8.

- Comparison of bacterial profiles to those of yeast and the NCI60 cell line panel.
- Parts of the description of Gram-level TSMs were removed.
- Figure 3 and description was moved to the SI (now Supplementary Fig. 2), now only shortly referenced in results and discussion each
- The description of Figure 4 (now 3) was simplified.
- The validation paragraphs were changed, among others the description of the DESI results (in response to comment 8) was moved to the SI and further validation sets were included, shifting focus away from specific markers and towards the workflow and its applicability as a whole.

10) PS: Please fix the numerous typos in the manuscript.

See comment 5).

Reviewer #3 (Remarks to the Author):

11) Using multiple MS technologies (REIMS, DESI, and MALDI) for feature comparison and as training datasets to identify TSMs, needs further validation. In the current form, it is not convincing that the biomarkers reported will be widely applicable to detect bacterial species in other clinical samples. In case the author wants to propose the workflow, I would suggest having more details on methodological validation and making computational tools/code publicly available for other researchers to reproduce the workflow.

Only REIMS has been used for database generation and as a training set to derive TSMs. DESI and LC-MS/MS have been used for *in situ* analysis. The high spectral similarity between DESI and REIMS has previously been documented.⁴ The second step of ion formation in DESI follows the same electrospray droplet evaporation mechanism as ESI, which is the standard ionization interface in LC-MS analysis⁵ and lipid ions observed are generally very similar.^{6,7} MALDI has not actually been part of this study for small metabolite detection, however, we will include it in our following discussion.

Based on our own experience and also several comparative studies published in literature, softer ionisation techniques, when deployed for *in situ* detection in negative ion mode, predominantly feature lipids and other small metabolites as singly deprotonated ions and for certain species such as ceramides as $[M+Cl]^-$. This has been described for REIMS^{8,9}, DESI^{6,10}, MALDI¹¹, or ESI ionisation sources from tissue extracts¹² with spectra generally dominated by intact phospholipid species within the m/z range of 600 to 1000.

A further preprint by Saharuka et al¹³, which is currently under revision by Nature Biotechnology that we have been involved with, compares spatial metabolomics protocols and technologies on a large scale and shows that the metabolite pool detected is generally very similar between DESI and MALDI, especially in negative ion mode.

To demonstrate these spectral similarities, we have included a comparison of bacterial profiles for *Bacteroides fragilis* recorded using REIMS and DESI. While relative abundances might change, in general, we can see the same species present in both modalities. This Figure is now located in the main manuscript as Figure 5.

Figure 5. Spectral comparison of *B. fragilis* using REIMS (a) and DESI-MS (b). TSMs are shaded in blue. Dark blue: *B. fragilis* specific markers. Light blue: TSMs for higher taxonomic levels. Their *m/z* values and taxa are (from left to right): 328.286 (*Bacteroidaceae*, family), 465.304 (*Bacteroidetes*, class), 555.520 (Isotope of 554.5156, *Bacteroidaceae*, family), 590.493 (*Bacteroidaceae*, family, followed by isotopic TSM peaks at 591.497, 592.490, 593.493, 594.496), 604.509 (*Bacteroidetes*, class, followed by isotopic TSM peaks at 605.513 and 606.505), 634.521 (*B. fragilis*, species), 677.524 (*B. fragilis*, species, followed by isotopic TSM peaks at 678.529 and 679.530), 752.542 (*B. fragilis*, species), 781.582 (Isotope peak of 780.5734, *B. fragilis*, species), 840.768 (*Bacteroidetes*, class).

We have followed the suggested refocusing of the manuscript, removed the data on specific samples using DESI and instead focused on the general applicability of the TSMs. We have introduced a new results and methods section on the detectability of the markers in complex samples, including several novel, independently recorded datasets.

Furthermore, the code for TSM generation and all original data are now available.

“Code availability. The code for TSM determination is available via the public GitHub repository <https://github.com/jsmckenzie/bacterialTSM>, which also includes worked examples for a subset of variables.

Data Availability. The 3274 bacterial isolates used to train and validate the TSM model are available at MetaboLights MTBLS10328. The DESI mass spectrometry imaging data of human colorectal tissues are available via MTBLS289. All DESI mass spectrometry imaging data from human and murine tissues can be accessed via MetaboLights MTBLS10846 and METASPACE (access link supplied in Supplementary Data 4). FISH results are uploaded in Figshare (https://figshare.com/articles/figure/_/26504344).”

12) Introduction: The author highlighted four challenges for the wide applicability of such methods in clinical samples. How this study addressed all four or any of those challenges is not clearly explained or well understood by the readers.

The 4 challenges mentioned by the reviewer are: 1) the variation of cell numbers in different clinical settings, 2) the large number of pathogens resulting in large genetic and metabolic diversity, 3) overestimated specificity of TSM markers if an underpowered microbial database is used, 4) the presence of other microbial species in many specimens.

With the current study, we have particularly aimed to address points 2 and 3, driven by the continued publication of undersized datasets comparing bacteria that differ on the highest

taxonomic levels or such datasets where markers were derived from a set of 2 closely related bacteria while ignoring all others. We wanted to assess whether the approach of a small molecular biomarker to detect bacteria is at all feasible or whether once the database becomes more complete, any specificity will get lost. We found that while the approach expectedly does not work for all bacterial species and taxa, it seems promising for some of the bacterial species, of which some have important clinical implications.

We have made the following changes to the objectives section in the introduction to address this comment:

“In this work, we address especially points i) and ii) assess the feasibility of a biomarker-based strategy that includes conserved small metabolite-based markers capable of identifying specific bacterial taxa (from phylum to species), which we refer to as taxon-specific markers (TSMs). 359 TSMs were found and subsequently deployed to identify pure bacterial cultures. We further touch upon points iii) and iv) through assessing whether these markers can be used to detect bacteria in situ in complex matrices.”

13) Abstract: Typo error at "Samsamples."

Typo was corrected.

14) Results: The data shown in Figures 1 to 3 are generated with REIMS, which is a different ionization technique than MALDI and DESI. Have the authors considered this aspect with direct comparison of features or transfer of TSMs?

This comment deals with a similar point to comment 11 to which we refer for changes made to the manuscript. Each TSM is a unique chemical entity that ultimately will need to be identified and then can be tracked in different polarities and as different adducts, ensuring transferability between methods. The spectra obtained from soft ionisation methods in negative ion mode however generally feature adducts of the type $[M-H]^-$ and occasionally $[M+Cl]^-$, which can be easily distinguished from each other in high-resolution mass spectrometry by the distinct Cl isotope pattern and spacing. Thus, we think that TSMs are sufficiently transferable within the scope of this study to enable detection across the mentioned modalities, as long as negative ion mode data is used.

15) Figure 1: It's hard to read the labels in the Sunburst plot. More description in legends is needed for the general scientific audience to understand the message of these plots. The interpretation of the plots is just to show the database composition without rationale or an overview of the selected database?

Why were m/z peaks from 50 to 200 Da excluded during analysis or not acquired?

In our opinion, it is imperative for the reader to see the database composition to ensure that the authors tried to cover microbial complexity adequately. We have amended the sunburst plot into a horizontal plot that contains the same information, but we hope is now easier to read and understand. We also restructured the corresponding paragraph to convey the aim of this figure better. The figure legend was also extended. Please see the new version below:

Figure 1. Overview training database composition. Horizontal stacked bar representation of order (a) and family level (b) of the database used to derive TSMs. The width of each bar is related to the percentage coverage of the database comprising 597 bacterial profiles of 232 bacterial species. Phylum level is indicated using brackets and colour (purple: Actinobacteria, blue: Bacteroidetes, cyan: Firmicutes, green-yellow: Fusobacteria, yellow: Proteobacteria). A detailed table on database composition can be found in Supplementary Data 1.

The REIMS database was acquired starting from m/z 150 instead of m/z 50. This is due to the mass spectrometer used for database acquisition (Thermo Exactive), which tends to skew mass spectral profiles if the mass range selected is too broad. As the primary focus was to record lipid profiles we thus chose to restrict the mass range to start from m/z 150 only. The mass range below m/z 200 was not chosen for a similar reason to that beyond 1000 Da, the overall spectral intensity in this range is very low, making it unlikely that a marker in this range will be detected in a complex matrix. This is illustrated in Figure R-1 below. We have amended the text in the manuscript to cover the additional range from 150-200 Da.

Figure R-1. Bar plot showing the cumulative intensity across m/z bins. The mean spectrum was determined from all 597 training set observations, first by individually interpolating each observation's spectrum to a fixed resolution of 0.001. After mean averaging, the intensities across each bin ($x \leq m/z < x+10$) were summed, as shown in the plot.

16) “5866 unique m/z values were detected over the entire training dataset.” This number of unique m/z is too large. Does it mean in each spectral data file? Out of how many m/z values in total?

We have changed this sentence to reflect that this is a total number of metabolites detected over the entire database. Seeing as many microbial metabolites are unique and not shared among the entire dataset (which was also the primary hypothesis underlying this study), we feel this number is appropriate to cover bacterial metabolism of 232 species over a mass range of m/z 200-1000. This number is further not deisotoped. The median number of peaks detected across the individual files was 985, and from these disparate lists of peaks, a total of 5866 variables were selected based on a minimum frequency of 10 in the smoothed plot.

“After spectral pre-processing, a total of 5866 unique m/z values were detected over the entire training dataset comprising metabolites from all 232 bacterial species.”

“**Pre-processing.** [...] Peak picking was performed on each mean spectrum using MATLAB’s *findpeaks* function with a noise threshold set to include peaks with a threshold greater than twice the spectral intensity’s 85th percentile; the median number of peaks detected across all files was 985.”

17) “359 features were found to be specific at a certain phylogenetic level and are considered TSMs.” It sounds like each TSM is with 359 features exactly associated with each phylogenetic level? Or are they totally unique features?

These 359 TSMs are unique features, distributed among the phylogenetic levels as shown in Figure 2c. We have changed this instance to clarify this fact, in combination with the expanded Materials and Methods section:

“Using a univariate approach, a total of 359/5866-features were found to be specific at a certain phylogenetic level (see Figure 2c) and are considered TSMs; the relevant criteria are explained in Materials and Methods.”

18) Figure 2: It mentions “phylum of each of the 597 observations,” which are again features. How do they arrive? Earlier, 359 features were considered to be TSMs.

Observations refer to the bacterial profiles in the database and is a term used in Matlab to describe the data structure. As this expression can be called bioinformatics jargon, we have removed it from the results and discussion sections of the manuscript and replaced it with “bacterial profiles in the training set” as well as clarified its meaning in the remaining instances.

19) Figure 2A) The scores for PC1 and PC2 are around 6%, which is a significantly low representation of variance across datasets.

We thank the author for spotting this. Having reviewed the code for plotting the figure, we noticed the eigenvalues had not been appropriately rescaled to express them as percentages. Having done so, the variance explained by components 1 and 2 has increased to 14.82% and 12.47%, respectively. We have updated Figure 2a accordingly:

20) "Of these, a total of 359 features were found to be specific at a certain phylogenetic level and are considered TSMs": Are these 359 TSMs not mentioned or described in Figure 2?

This sentence has been highlighted already in comment 17. We have added a reference to Figure 2c in the revised manuscript version.

“Using a univariate approach, a total of 359/5866-features were found to be specific at a certain phylogenetic level (see Figure 2c) and are considered TSMs; the relevant criteria are explained in Materials and Methods.”

21) What does the TSM across m/z distribution show? The AUC specificity indicated by circle size seems relatively weak.

In this plot (Figure 2b) there is a partial overlap with the plot in Figure 2c. However, we feel that these plots still merit being shown alongside each other. b) shows the spread of TSMs over the m/z dimension while giving an idea of the specificity of each marker, c) makes it clearer for which phylogenetic level how many markers are found and from which phylum of bacteria they originate.

In b), the smallest circle size is representing an AUC of 0.8. As detailed in the manuscript, the median AUC of all 359 markers is 0.9983, and the minimum is 0.8781. The AUC circle sizes from 0.80-1.00 were selected based on the TPR threshold of 0.80. Currently the database is still comparatively small, comprising only a handful of most species, and as the species ID in the database has not been confirmed beyond MALDI-MS identification, the occasional misclassified species is expected. Thus, a too stringent AUC value will result in the (potentially unwarranted) loss of a number of TSMs.

22) Since the majority (>80%) of TSM markers are found for the members of the phylum Bacteroidetes, does it skew the further data analysis towards a particular phylum or species?

Each marker is determined independently from all others, and whilst most were detected for members of the *Bacteroidetes* phylum (202/359 = 56%), there is no inherent bias towards this or any other group in subsequent analysis. *Bacteroidetes* is not over-represented in the dataset, comprising 50/597 observations (< 10%). Once determined, each marker is considered as an individual entity as part of the validation process.

The likely reason for the higher number of TSMs is related to the mass spectrometric profiles being sufficiently different; from this it seems that the bacterial metabolism is also sufficiently different. We have changed the manuscript as follows to clarify that the number of markers is not caused by a skewed database composition:

“As Figure 2c (and Supplementary Fig. 1 as a more detailed breakdown) shows, most of these markers are found for the members of the phylum Bacteroidetes, although they only constitute <10% of database spectral profiles (see Figure 1), suggesting a distinctly different metabolism in this phylum of bacteria.”

23) “Of the 116 species-specific TSMs, 35 are identified as isotope peaks,” which means 30% of total peaks are isotopic. I am not sure if this will not affect the downstream predictions and data analysis.

Isotope peaks in this context are predominantly ^{13}C isotopes as well as in some cases ^{37}Cl peaks and their mixed forms. We have made a conscious decision to include these isotopes into the list of potential TSMs for two main reasons: i) they act as internal sanity check as the same result should be expected for the peak itself and its isotope; ii) in some cases, the monoisotopic peak might experience overlap (in the bacterial database) while the isotopic peak has no interference, this has been observed in 13 of these 35 cases. Ultimately, interference might also be an issue in complex matrices, which will enable the use of the alternative marker.

We have amended the specific paragraph as follows, please note that the number of peaks identified as isotopes has increased to 43 (see Supplementary Data 2):

“Of the 116 species-specific TSMs, 43 are identified as isotope peaks. We decided to include these as they can serve for corroborative purposes or as alternative markers in cases where there may be spectral interference in either the complex matrix or the training dataset. For instance, in 15 of these cases, the original peak is not among the TSM marker list, possibly due to overlap with spectral profiles of other bacterial species.”

23) Lipid identifications need more confirmation at Level-1 whenever possible to improve the reliability of reported lipid biomarkers: Just using exact mass search on such datasets will not give reliable IDs or even tentative IDs. Consider using MS/MS data for IDs.

Due to the variety of bacterial species (1400 pathogenic species relevant for clinical practice alone) and the huge genetic and metabolic differences between these species as compared to mammals, often very little literature is available on bacterial metabolites in a very specific bacterial species. While there are a few species that are comparatively well characterized due to their clinical relevance (*P. aeruginosa*, *S. aureus*, *C. difficile*, *Mycobacteria*) or their use as model organisms (*B. subtilis*, *E. coli*, *C. albicans*), this is not the case for the majority of bacteria. MicrobeMASST, as outlined in the discussion might be a good next step to advance these microbial metabolite identification, however, at this point, MS/MS data in there can not be freely browsed and are in positive polarity only.

We have attempted to clarify the level of our lipid annotations and avoid misleading lipid IDs. Level 1 identifications (according to Schymanski *et al.*¹⁴) unfortunately require comparison with reference standards, which in many cases will not be available. At this point, identifications up to Level 3 or 2 the most will be achievable through the interpretation of MS/MS data and comparison with literature reports. Thus, to be transparent about the quality of our annotations, we have updated our TSM marker list (Supplementary Data 3) with the identification confidence level as described by Schymanski *et al.*

24) Please describe how m/z 604.509 is identified as ceramide 35:0;O3, as exact mass or adduct masses are not matching.

m/z 604.509 can be identified as the $[M + Cl]^-$ adduct ion of ceramide 35:0;O3 with a mass deviation of 0.0012 Da. The $[M + Cl]^-$ adduct ion is indicated by the presence of an isotope peak at m/z 1.997 Da and 30% relative intensity (see Figure R-2b). This has however not been clear from the original text or the supplementary table for that matter, for which we apologise. We have amended the identification as chloride adducts (Supplementary Data 3) to make this clearer. To assign this particular molecular species, we have internal REIMS MS/MS data, and comparison with literature reports in closely related bacterial species.

This dihydroceramide (DHC) was recently detected in *B. thetaiotaomicron* as both $[M-H]^-$ and $[M+Cl]^-$ by Frankfater *et al.* with HR ESI-MS¹⁵ (Fig S1e from Frankfater *et al.*¹⁵ is attached below as Fig. R-2a, authors state to have confirmed ID using MS/MS on $[M-H]^-$ analogue). Our own MS/MS spectrum for m/z 604 is shown below in Fig. R-2b (acquired on a Q-TOF instrument) and shows loss of the Cl^- ion resulting in the formation of a $[M-H]^-$ ion at m/z 568.

Ref.¹⁵ has shown the process of structure identification of a homologue DHC at m/z 554, defining the structure of d17:0/ β h17:0-Cer, which is one CH_2 shorter than the one at m/z 568. The MS2 and the MS3 can be seen in the Fig. R-2c and d. The MS2 was dominated by the ion of m/z 328, arising from the loss of the β -OH 17:0-fatty acid substituent as a 15:0-aldehyde by β -cleavage (226 Da, see Fig. R-2d), corresponding to the ion of m/z 342 in our MS2 (Fig. R-2b). Additionally, our MS2 contained a fragment ion of m/z 241 from the fatty alcohol, suggesting a structure of d18:0/ β h17:0-Cer.

Figure R-2. a) Overview spectrum from ref.¹⁵ showing compounds identified as dihydroceramides (DHCs) as $[M-H]^-$ (labelled blue) and $[M+Cl]^-$ (red) adducts and b) MS/MS spectrum acquired from m/z 604.5 $[M+Cl]^-$ from *Bacteroides fragilis* in this study. c) MS2 spectrum of the $[M-H]^-$ ion of m/z 554 and d) its MS3 spectrum of the ion of m/z 328 (554 \rightarrow 328) from ref.¹⁵, defining the structure of d17:0/ β h17:0-Cer, a homolog to the DHC at m/z 568 with CH_2 (14 Da) differences. The fragmentation mechanism is further shown in d).

In addition to different isotopes now being more clearly marked in the manuscript and supplementary material, we have clarified the presence of this peak as [M+Cl]⁻ adduct in the text and added the references to support this assignment:

“TSM with m/z 604.509 (Figure 3d) is specific for the class of *Bacteroides* and assigned as ceramide 35:0;O3 as [M+Cl]⁻ adduct.^{46,47}”

25) What does this statement mean? Is all datasets cross-verified for IDs with MALDI-MS for REIMS datasets? “All files used in this study were identified with MALDI-MS (using a Bruker Biotyper).”

The database entries were acquired from >3000 individual clinical isolates which were isolated during routine clinical practice at the Clinical Microbiology Department at Charing Cross Hospital, London. Once a blood culture bottle flags positive, it is plated onto solid culturing medium and the bacteria grown are then identified to species-level using the Bruker Biotyper and experienced microbiology personnel using ribosomal protein patterns and matching to database spectra. Subsequently, in line with guidelines, a backup stock is generated and stored at -80 degrees for up to 12 months. The isolates used in this study were re-cultured from these backup stocks. The assumed ID of each isolate is the one provided by the routine clinical microbiology workflow outlined above. To clarify, we have amended the mentioned sentence as follows:

“All bacteria were isolated during routine practice at the NWLP clinical microbiology laboratory at Charing Cross Hospital, UK. Bacterial IDs were obtained using the Bruker Biotyper platform (identification through ribosomal protein profiles using MALDI-MS) by Biomedical Scientist microbiology staff during routine clinical microbiology practice following manufacturers protocols (expected misclassification rate 2-5%).⁶²”

26) Similar descriptions for the ID of m/z 949.777 are required. Is there any reference or database citation available for the identification of “C15:0 substituted phosphoglycerol dihydroceramide”? There is only one (ref. 47) mentioned for these identifications in *Porphyromonas gingivalis*. However, the reference does not report exact masses or any other standard identification procedures.

We again have to apologise for not having been very clear here. This specific TSM was identified as an isotopic peak of a C15:0 substituted PG Dihydroceramide at m/z 946 (Fig. R-4a). In our study, the m/z 946 and its isotopes at 947, 948 and 949 are all considered TSMs for the family *Porphyromonadaceae*. To use an isotope peak in this Figure was probably not a good choice from our side. The respective set of plots with the original peak would look as shown below (Figure R3).

Figure R-3. TSM abundance over several members of the Bacteroidetes class members (Previously Figure 4 panels c-f).

To address the assignment of this mentioned compound in particular, our own REIMS-MS2 spectrum of m/z 946 in *Parabacteroides distonasis* is shown in Fig R-4b; the signature PG head group fragment at m/z 171 was captured. Tavana *et al.* have also detected this m/z 946 peak in *Porphyromonas gingivalis* (which belongs to the same family *Porphyromonadaceae*, so is closely related to *P. distonasis*), with a similar MS2 spectrum (Fig. R-4c).^{16,17} Nichols *et al.* (above-mentioned reference 47¹⁸) have identified this C15:0 substituted PG dihydroglycerolamides at m/z 946 together with its homologues at m/z 932 and 960 from *P. gingivalis* with the additional help of NMR. Their MS and MS2 spectra are shown in Fig R-4d, their MS2 of m/z 946 matching well with ours (Fig. R-4b). While the reference does indeed not mention exact masses, it was showing molecular structures. These allowed us to calculate the exact masses ourselves and perform assignment through exact mass and the comparison of MS/MS spectra.

As outlined also as part of the next comment, bacterial metabolism is characterised very poorly for the vast majority of bacterial species, which makes it impossible to find reports for exactly the same compound in exactly the same bacterial species at all times. In this case, a spectrum from a member of the same bacterial family that this TSM was assigned to had to be resorted to.

Figure R-4. a) Zoom into the full scan spectrum at peak m/z 946 and its isotopes from *Parabacteroides distasonis* acquired by REIMS, and b) the corresponding MS2 of the monoisotopic peak at m/z 946. c) MS2 of m/z 946 from the extraction of polar lipids of *Porphyromonas gingivalis* from reference ¹⁶. d) Negative MS of the HPLC fraction of *P. gingivalis* dominated by dihydroglycerolamides, whose presences and structure were confirmed by NMR analyses from reference ¹⁷. e) -g) MS/MS of the 960, 946, and 932 m/z homologue parent phosphoceramide of *P. gingivalis* from Ref.¹⁷. The daughter ions at m/z 171 recovered for these three parent ions represent the phosphoglycerol head group, structure shows in f); e) proposed phosphoceramide structure for the ion at m/z 690; g) proposed phosphoceramide structure for the 718 m/z ion.

We have however decided for the manuscript to merge this Figure 4 with Figure S2 on C. *difficile* that was previously shown in the SI as part of refocusing the manuscript on the TSM workflow and improving manuscript clarity (following comments from especially reviewer 2). The novel Figure is shown below:

Figure 2. Assessment of the quantitative nature of TSMs. **a** PCA scores plot of data recorded for *Clostridium difficile* cultured on 5 different culturing media (Braziers: *C. difficile* selective medium, CBA: Columbia blood agar, CHOC: Chocolate agar, AZT: CBA with aztreonam [8 mg/L] for fastidious Gram-positives, NEO: FAA with Neomycin for fastidious anaerobes). **b** Normalised (TIC and log transformation) intensities of TSMs found for *C. difficile* over the different culturing media. **c** PCA scores plot of data recorded for the *Bacteroides* phylum including taxonomic relationship. The colour code is shown beside the list of species in **d**. **d** and **e**: Abundance of TSMs within species of *Bacteroides* phylum for class *Bacteroidetes* specific marker at *m/z* 604.509 and class *Flavobacteria* specific marker at *m/z* 640.537, respectively.

We would also like to refer back to comment 24 and our attempts to make any isotope and adduct assignments more clear in this revised manuscript version.

27) The author is missing a critical metabolite identification process here without any justification: “As for most TSMs currently, no fragmentation data is available for higher-level identification and comparisons for this proof of principle are made based on exact mass only.” It is fairly straightforward to perform MS/MS with standard LC-MS analysis and on-tissue MS/MS with MALDI and DESI.

This comment shows similarities to comment 23, dealing with the identification confidence level of our analytes. To be transparent we have now extended our TSM marker table with the identification confidence level for each metabolite and we have put more effort into increasing the level for the markers to our best abilities within the scope of this revision (Supplementary Data 3).

While on tissue MS/MS can theoretically be performed to confirm the same marker is detected, this requires instrumentation capable of fragmentation analysis (the Thermo Exactive classic used to record the DESI-MSI and REIMS dataset is not) and equally important, sufficient sensitivity to perform fragmentation analysis, which is frequently problematic due to the abundant matrix effects in imaging analysis.

The proposed approach will certainly be stronger and more versatile if all peaks are identified as this will enable the analysis in positive ion mode and LC-MS with adducts resulting from mobile buffer systems through calculations of other adducts. It will also be critical for the development of targeted and quantitative approaches.

Unfortunately, in the absence of authentic standards or even tentative identification for many of the markers we detected, it is highly unlikely that an approach deploying MS/MS alone will enable the identification of the compounds in question. Lipids also frequently show different fatty acid compositions, which will complicate differentiation between different possibilities further. We do agree that identification is one of the necessary next steps towards the development of clinical applications and we have included the following passages in the discussion section to reflect this:

Methods section:

“Using exact mass, isotopic patterns, database search (The Human Microbial Metabolome Database⁴¹, *Pseudomonas aeruginosa* metabolite database⁴², LIPID MAPS⁴³) and literature search, we could tentatively identify 177 of the overall 359 TSMs which are listed in Supplementary Data 3; no possible annotations were found for the remaining TSMs, which is associated with the only poorly characterised metabolic diversity among bacteria.^{44 45} While not critically important for assessing the feasibility of bacterial small molecular biomarkers in general, precise metabolite annotation will eventually facilitate the translation of TSMs to other MS modalities through enabling the calculation of adduct exact masses under different experimental conditions and polarities.”

Discussion section:

“To further promote the clinical translation of these markers, the identification of the molecular ID of each marker is a critical step in future TSM validation. Due to the high genetic variability of microbial species, the molecular identity of many TSMs is currently unknown or only tentatively assigned and remains to be confirmed. Knowing the metabolite in question will enable a more direct transfer of TSMs between different MS modalities and polarities through the tracking of different potential adducts. In this study we have performed our analysis in negative ion mode only (adducts predominantly of the type $[M-H]^-$ and $[M+Cl]^-$ which can be easily distinguished based on isotopic patterns) and as a result different adduct formation behaviour was not taken into account. Due to the different chemical nature of the different TSMs, the exact ionisation behaviour and thus sensitivity in a clinical application for each marker will depend on its physicochemical properties and the matrix in question.

[...] Recently Zuffa *et al.* have developed an MS/MS-based microbial metabolite search platform, MicrobeMASST,⁴⁵ which allows users to search for known/unknown MS/MS profiles against public databases of microbial single cultures and map the results to taxonomic trees. Joint use with such platforms might further improve the identification and reliability of the TSMs.

28) The justification in the Discussion section is not convincing. “The presented approach was bioinformatically adjusted to allow the occasional misidentification.”

Bacterial identification using MALDI-MS is expected to generate a wrong ID in 2-5% of the cases, although predominantly on similar taxonomic levels (family to species). Thus, using too strict rules for biomarker selection from the database files might result in the exclusion of otherwise valid markers. To avoid further confusion, we have removed this paragraph and added the expected misclassification rate in the methods section for REIMS. In accordance with other comments, the data processing pipeline was further expanded and the data and code were deposited in public repositories.

29) Is the data submitted mostly acquired in negative mode? Is this the case with all other datasets?

Indeed, only negative ion mode data was used in this study. Empirically, negative ion mode is known to yield better classification results due to a higher number of lipid classes detected in negative over positive polarity. While some markers are expected to be detected in positive mode too, it is fair to expect that this is not the case for all markers. We have clarified the ion mode used in the methods section under REIMS and external validation and discussed the transferability of markers between methodologies and polarities in the Discussion section:

“[...] Knowing the metabolite in question will enable a more direct transfer of TSMs between different MS modalities and polarities through the tracking of different potential adducts. In this study we have performed our analysis in negative ion mode only (adducts predominantly of the type $[M-H]^-$ and $[M+Cl]^-$ which can be easily distinguished based on isotopic patterns) and as a result different adduct formation behaviour was not taken into account. Due to the different chemical nature of the different TSMs, the exact ionisation behaviour and thus sensitivity in a clinical application for each marker will depend on its physicochemical properties and the matrix in question.”

30) Validation using independent data:

How can splitting the same data be qualified as “independent data,” as mentioned in “Of the 3274 files forming part of the bacterial database, the remaining 2677 files not included in the training set were used as a test set for validation”? The independent datasets should be acquired or originated independently. What were the criteria for this split of data files? Overall, the validation part is weak and needs more description and data analysis.

We have improved our validation procedure in line with previous comments 1, 7, and 11. The independent data referred to in comment 31 (and also comment 7) refers to spectral profiles from independent clinical isolates, recorded by the same operator on the same instrumental platform over a period of 3 years.

The split of data into training and validation set (validation on the level of being able to identify a pure bacterial culture) was performed as outlined (now in more detail) in Materials and Methods (compare comment 3) to produce a training dataset with 3-7 files per bacterial species. If less than 3 files were present in the overall dataset of 3274 files, all were included in the training database, otherwise files to be included were randomly chosen. The whole database is publicly accessible at MetaboLights (identifier MTLBS10328), the files used in the training dataset are clearly distinguishable in Supplementary Data 1.

We have now expanded this section with additional data from a second, different instrumental platform recorded by another operator (see comment 1 and novel Figure 4) as well as further imaging data complemented by imaging-based validation (see comment 33).

31) Feasibility of detecting TSM in colorectal cancer specimens:

44 tumors were analysed with DESI-MS, which is again a different ionization method. How are the previous datasets, TSMs, and models derived from different platforms transferable?

The goal of using different ionisation methods was to show the MS method-agnostic nature of the TSMs. We have included a section in the manuscript that discusses similar data output observed for REIMS, DESI and LC-MS as well as Figure 5 showing a spectrum for *B. fragilis* recorded with REIMS and DESI each. The spectral profiles obtained are remarkably similar, especially in negative ion mode, where peaks are largely of the type $[M-H]^-$. This has also

been discussed in response to comment 11 and 14, which we would like to additionally refer to.

32) Figure 6: Particular m/z from TSMs were associated with bacteria in tumors. Other than the H&E image showing necrotic area, there should be other imaging such as IHC or bacteria-specific Immunofluorescence stained

This was unfortunately not possible for the initial CRC study. Thus, we have recorded several new datasets of human and murine origin using DESI-MSI (new operator, Thermo Q-Exactive instrument) and subjected these to fluorescence *in situ* hybridisation (FISH) using a general pan-bacterial EUB probe to visualise bacteria (in collaboration with Manuel Liebeke). These results are now included in the main paper (Figure 6) and show that indeed bacterial hotspots are seen in areas observed to contain bacteria based on our TSM analysis. The added passages are shown below:

Figure 3. Comparison between MSI and FISH images. a-d: Murine stomach tissue, one week after infection with *Helicobacter pylori*. **a** the overlaid DESI-MSI image for two tissue markers and a TSM of *H.pylori* at m/z 299.260. MS spatial resolution: 75 μm . **b** the FISH image of the same tissue section from **a** recorded after MSI analysis. Objective: 40x. Blue: DAPI probe for DNA. Red: Eub338 DOPE bacteria probe. **c** and **d** Zoomed FISH images, regions were boxed by white rectangles in **a** and **b**. 40x objective. **e-f** Tumour tissue of a mouse model for colorectal cancer (*Apc* 1638N/wt x *pvillin-Kras* V12G)⁵⁶. **e** Overlaid DESI-MSI images for three different TSMs of class *Bacteroidetes*. MS spatial resolution: 75 μm . **f** FISH image of neighbouring tissue section to **e**. 40x objective. Blue: DAPI probe for DNA. Red: Eub338 DOPE bacteria probe. Green: CF319a probe for phylum *Bacteroidetes*, covering 90% *Flavobacteria*, 90% *Sphingobacteria* and 30% *Bacteroides*.¹⁹ Mass tolerance for generation of the MS images: 10 ppm. See Supplementary Fig. 7 for H&E staining images. An example of a murine jejunum tissue can be found in Supplementary Fig. 8.

“To assess whether the bacterial hotspots detected using TSMs correlate to sites of actual bacterial abundance, we have analysed tissues from a murine colorectal cancer model as well as a murine *H. pylori* gastric infection model using spatial metabolomics while a neighbouring section was subjected to fluorescence *in situ* hybridisation (FISH) analysis. Figure 6a-d shows murine gastric tissue 1 week post infection with *H. pylori*. The marker m/z 299.260 (*Helicobacteriaceae* family), possibly belonging to 11-MeO-heptanoic acid⁵⁶ was detected in a distinct hotspot pattern. Subjecting the neighbouring section to FISH shows hotspots of bacterial presence at the same locations. As not many bacteria are capable of surviving in the harsh gastric environment, it is thus likely that these are indeed originating from *H. pylori*.

A more complex picture can be seen in murine colorectal cancer tissue shown in Figure 6e-f. Complex microbial communities are expected in colorectal tissue.^{36,51} Several markers belonging to the *Bacteroidetes* class, abundant intestinal microorganisms, are observed in this tissue, similar to those observed in human colorectal tissues (see Supplementary Fig. 3-5). TSM, as well as FISH imaging reveal a pattern of microorganisms locating to hotspots. Discrepancies are expected, as the FISH probe coverage and TSM coverage are not identical. However, the pan-bacterial Eub338 probe should be detected everywhere where TSMs are detected. While allowing for some differences in localisation due to non-identical tissue sections, this seems to be the case. Differences in the intensity of markers between FISH and TSMs might be related to actual local bacterial composition as well as differences in ionisation response between the different markers.”

33) Why was LC-MS analysis not performed on tumor samples itself for confirmation of identification? What is the link between 48 fecal samples and tumor samples or training datasets?

We have not performed LC-MS on the tumour samples itself as this was planned as an independent MS imaging study with its own study objectives. Due to the hotspot pattern observed for most TSMs, it might further be the case, that TSMs are not found in the LC-MS data due to dilution effects when homogenising heterogeneous tissues.

We have chosen this dataset for assessing the detectability of TSMs due to its availability on METASPACE, its tissue of origin (colon) and matching ion mode and mass range. While both sample sets originate from the colon and have been collected at Imperial College London under the study lead of Dr. James Kinross and Prof. Zoltan Takats, there is no link between the 48 faecal samples and the 44 imaging samples, which originate from independent studies on different donor cohorts with independent study objectives. Metagenomic sequencing was performed on both specimens, and proved on order and class level that the respective groups of bacteria are present. The summary data is added in the Supplementary Data 5.

To clarify the relationship (or absence thereof) between the different validation datasets, we have added a new table into the supplementary material detailing all samples used for validation, their lab of origin, time frames and instruments used for recording etc (Supplementary Table 3). We have further changed the Materials and Methods section on validation, dividing it into validation on pure bacterial cultures and the detectability of TSMs in complex samples.

References:

- 1 Bregy, L. *et al.* Differentiation of oral bacteria in *in vitro* cultures and human saliva by secondary electrospray ionization – mass spectrometry. *Scientific Reports* **5**, 15163 (2015). <https://doi.org/10.1038/srep15163>

- 2 Povilaitis, S. C. *et al.* Identifying Clinically Relevant Bacteria Directly from Culture and Clinical Samples with a Handheld Mass Spectrometry Probe. *Clinical Chemistry* **68**, 1459-1470 (2022). <https://doi.org:10.1093/clinchem/hvac147>
- 3 Bardin, E. E. *et al.* Metabolic Phenotyping and Strain Characterisation of *Pseudomonas aeruginosa* Isolates from Cystic Fibrosis Patients Using Rapid Evaporative Ionisation Mass Spectrometry. *Scientific Reports* **8**, 10952 (2018). <https://doi.org:10.1038/s41598-018-28665-7>
- 4 Golf, O. *et al.* XMS: Cross-Platform Normalization Method for Multimodal Mass Spectrometric Tissue Profiling. *Journal of the American Society for Mass Spectrometry* **26**, 44-54 (2015). <https://doi.org:10.1007/s13361-014-0997-6>
- 5 Honarvar, E. & Venter, A. R. Comparing the Effects of Additives on Protein Analysis Between Desorption Electrospray (DESI) and Electrospray Ionization (ESI). *Journal of The American Society for Mass Spectrometry* **29**, 2443-2455 (2018). <https://doi.org:10.1007/s13361-018-2058-z>
- 6 Zhang, J. I. *et al.* Rapid direct lipid profiling of bacteria using desorption electrospray ionization mass spectrometry. *International Journal of Mass Spectrometry* **301**, 37-44 (2011).
- 7 Abbassi-Ghadi, N. *et al.* A Comparison of DESI-MS and LC-MS for the Lipidomic Profiling of Human Cancer Tissue. *Journal of the American Society for Mass Spectrometry* **27**, 255-264 (2016). <https://doi.org:10.1007/s13361-015-1278-8>
- 8 Golf, O. *et al.* Rapid evaporative ionization mass spectrometry imaging platform for direct mapping from bulk tissue and bacterial growth media. *Anal Chem* **87**, 2527-2534 (2015). <https://doi.org:10.1021/ac5046752>
- 9 Paxton, T. in *Ambient Ionization Mass Spectrometry in Life Sciences* (ed Kei Zaitzu) 241-270 (Elsevier, 2020).
- 10 Pruski, P. *et al.* Direct on-swab metabolic profiling of vaginal microbiome host interactions during pregnancy and preterm birth. *Nature Communications* **12**, 5967 (2021). <https://doi.org:10.1038/s41467-021-26215-w>
- 11 Engel, K. M. *et al.* A new update of MALDI-TOF mass spectrometry in lipid research. *Progress in Lipid Research* **86**, 101145 (2022). <https://doi.org:https://doi.org/10.1016/j.plipres.2021.101145>
- 12 De Vijlder, T. *et al.* A tutorial in small molecule identification via electrospray ionization-mass spectrometry: The practical art of structural elucidation. *Mass Spectrom Rev* **37**, 607-629 (2018). <https://doi.org:10.1002/mas.21551>
- 13 Saharuka, V. *et al.* Large-Scale Evaluation of Spatial Metabolomics Protocols and Technologies. *bioRxiv*, 2024.2001.2029.577354 (2024). <https://doi.org:10.1101/2024.01.29.577354>
- 14 Schymanski, E. L. *et al.* Identifying Small Molecules via High Resolution Mass Spectrometry: Communicating Confidence. *Environmental Science & Technology* **48**, 2097-2098 (2014). <https://doi.org:10.1021/es5002105>
- 15 Frankfater, C. F., Sartorio, M. G., Valguarnera, E., Feldman, M. F. & Hsu, F.-F. Lipidome of the *Bacteroides* Genus Containing New Peptidolipid and Sphingolipid Families Revealed by Multiple-Stage Mass Spectrometry. *Biochemistry* **62**, 1160-1180 (2023). <https://doi.org:10.1021/acs.biochem.2c00664>
- 16 Tavana, A. M. *et al.* Phospholipid analogues of *Porphyromonas gingivalis*. *Journal of Applied Microbiology* **88**, 791-799 (2000). <https://doi.org:https://doi.org/10.1046/j.1365-2672.2000.01026.x>
- 17 Nichols, F. C. *et al.* Structures and biological activity of phosphorylated dihydroceramides of *Porphyromonas gingivalis*. *Journal of Lipid Research* **45**, 2317-2330 (2004). <https://doi.org:10.1194/jlr.M400278-JLR200>
<https://doi.org:10.1038/nrmicro1888>

REVIEWERS' COMMENTS

Reviewer #1

This is a revision of a previously submitted manuscript. Again, this reflects a great deal of work in an area that does hold promise but definitive validated results are important for the future.

Concerns:

1) The authors have answered this reviewer's questions, but for example, on bacterial quantification, the answer is not direct and does meander. The answer is that it is not possible at this time but with more experience and more validation of markers it may be possible in the future. Because of this approach, this reviewer is not convince the manuscript is better. By way of answering the questions the manuscript appears more disjoint and possibly harder for the uninitiated reader to understand. Most readers do not know a lot about metabolomics and this needs to be considered when writing the manuscript. This is mostly stylistic.

To make Metabolomics easy to understand we have added an introductory statement on metabolomics in our introduction section.

“Metabolomics is the comprehensive study of small molecules (metabolites), within cells, bio-fluids, tissues, or organisms. These metabolites are the final downstream products of cellular processes and provide a snapshot of the physiological state of an organism at a given time. By analyzing metabolomic patterns, we can identify specific biomarkers that differentiate between biological states, making metabolomics a powerful tool for microbial identification and understanding host-pathogen interactions.¹⁴ [...] “

We have overworked the Discussion section on bacterial quantification, however, we believe that ultimately a validation on individual marker level will be required with regards to very specific application scenarios (matrices) which cannot be delivered as part of this study.

2) Where is figure 4 in the manuscript. It goes from Fig. 3 To Fig. 5.

We apologize for this error, the numbering of the pictures has now been corrected.

3) Line 275. The sentence is not complete.

“Tissue specimens were either from a tumour lesion or 5 to 10 cm distant.” was changed to “Tissue specimens were either from a tumour lesion or 5 to 10 cm distant to the lesion.”

4) Line 283. What does the label (A15 10cm S3) represent? Please explain.

We apologise for the lack of explanation. We have modified this sentence in the manuscript:

“No bacterial markers were detected in 1/44 samples (A15 10cm S3, full sample list can be found in Supplementary Data 4).“

A detailed explanation has been added to the legend of Supplementary Data 4:

“Supplementary Data 4. Metaspace and MetaboLights access links for imaging datasets.

For human colorectal tissue samples, patients underwent hemicolectomy and were sampled from the centre of the tumour (CT) and 5 and 10 cm away from the tumour (regarded as healthy).“

We added one [Tissue Site] column in Supplementary Data 4, indicating where the tissue sample was taken from.

Reviewer #3:

The authors have considerable efforts in revising the manuscript by including additional validation, comparison of FISH and MSI images and explaining metabolite identification and confidence levels.

It will be useful to include some more details (which authors explained in ruttal such as FISH and MSI comparison, data analysis methods and lipid identification interpretation) directly into the manuscript or supplementary information. These details may not be relevant to main conclusions but can be valuable to working researchers in the field.

We have carefully overworked all Methods sections in both the manuscript and the Supplementary Material and added more details.

This is especially the case for the Validation of markers against pure bacterial cultures, 16S rRNA Fluorescence In Situ Hybridization (FISH) analysis and the Statistics and Reproducibility section. Data and Code availability statements were included and all materials are made freely available (although some are still under curation at the MetaboLights repository).